# Random Policy Valuation is Enough for LLM Reasoning with Verifiable Rewards

**Haoran He**[1][*]  **Yuxiao Ye**[1][*]  **Qingpeng Cai**[2]  **Chen Hu**[3]  **Binxing Jiao**[3]
**Daxin Jiang**[3]  **Ling Pan**[1][†]

[1]Hong Kong University of Science and Technology [2]Kuaishou Technology [3]StepFun
`haoran.he@connect.ust.hk, lingpan@ust.hk`

## Abstract

RL with Verifiable Rewards (RLVR) has emerged as a promising paradigm for improving the reasoning abilities of large language models (LLMs). Current methods rely primarily on policy optimization frameworks like PPO and GRPO, which follow generalized policy iteration that alternates between evaluating the current policy's value and improving the policy based on evaluation. While effective, they often suffer from training instability and diversity collapse, requiring complex heuristic tricks and careful tuning. We observe that standard RLVR in math reasoning can be formalized as a specialized finite-horizon Markov Decision Process with deterministic state transitions, tree-structured dynamics, and binary terminal rewards. Though large in scale, the underlying structure is simpler than general-purpose control settings for which popular RL algorithms (e.g., PPO) were developed, suggesting that several sophisticated techniques in existing methods may be reduced or even omitted. Based on this insight, we prove a surprising result: the optimal action can be recovered from the Q-function of a fixed uniformly random policy, thereby bypassing the generalized policy iteration loop and its associated heuristics. We introduce **R**andom **P**olicy **V**aluation for Div**e**rse **R**easoning (ROVER) to translate this principle into a practical and scalable algorithm for LLM math reasoning, a minimalist yet highly effective RL method that samples actions from a softmax over these uniform-policy Q-values. ROVER preserves diversity throughout training, allowing sustained exploration of multiple valid pathways. Across multiple base models and standard math reasoning benchmarks, ROVER demonstrates superior performance in both **quality** (**+8.2** on pass@1, **+16.8** on pass@256) and **diversity** (**+20.5%**), despite its radical simplification compared to strong, complicated existing methods.

*"Simplicity is the ultimate sophistication." - Leonardo da Vinci*

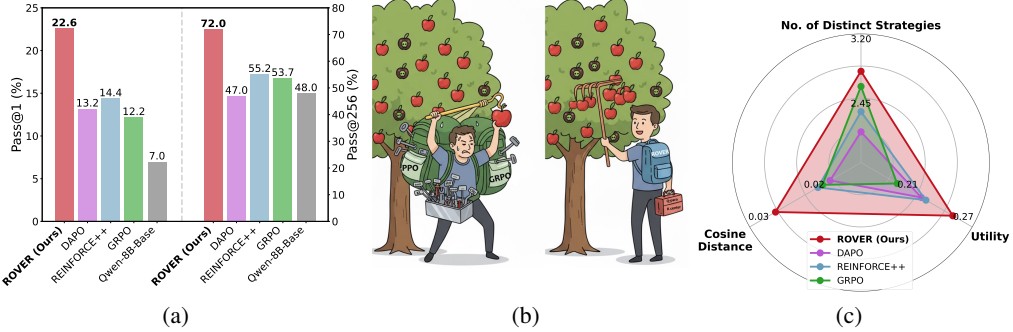

Figure 1: (a) Pass@1 & Pass@256 results on Qwen3-8B-Base averaged over AIME24, AIME25, and HMMT25 tasks. (b) Illustrative example demonstrating that ROVER achieves high-quality solutions with a lightweight procedure while maintaining diversity. (c) Comparison of different methods on multiple diversity metrics. Higher value denotes better diversity.

---

[*]Equal contribution.  [†]Corresponding author.

# 1 INTRODUCTION

RLVR has emerged as a promising paradigm for post-training LLMs and enhancing reasoning capabilities (Jaech et al., 2024; Guo et al., 2025). The field has primarily relied on Proximal Policy Optimization (PPO) (Schulman et al., 2017), a powerful algorithm originally designed for standard deep RL benchmarks such as computer games and robotic control. This general-purpose algorithm and its specialized derivatives like Group-Relative Policy Optimization (GRPO) (Shao et al., 2024) have achieved notable successes in improving LLM reasoning performance. Fundamentally, current methods follow the generalized policy iteration (GPI) (Sutton et al., 1998) paradigm, which iteratively alternates between *evaluating* the current policy and *improving* it based on the evaluation.

Despite its success, they suffer from unstable learning dynamics (Yang et al., 2025a) and entropy collapse (Huang et al., 2024; Yang & Holtzman, 2025) induced by the reward-maximizing nature within the *iterative policy evaluation-improvement cycle*. As the policy continuously evolves, the evaluation target becomes non-stationary, leading to training instability and narrowed exploration spaces. Recent variants mitigate this through an intricate ballet of heuristic techniques such as clipping (Yu et al., 2025), KL regularization (Liu et al., 2025a), and data selection (Liang et al., 2025). While incorporating these tricks offers partial improvements, they add layers of implementation complexity and typically require careful, case-specific tuning (Liu et al., 2025d).

We take a fundamentally different approach by examining the underlying structure of LLM math reasoning tasks with verifiable rewards. Unlike standard RL environments that sophisticated RL algorithms like PPO were originally designed for and evaluated (e.g., discrete computer games with cyclic state transitions that forms a graph instead of a tree (Bengio et al., 2021), robotics with continuous spaces, possibly with stochastic transitions and intermediate rewards), standard RLVR for math reasoning corresponds to a specialized finite-horizon Markov Decision Process (MDP) with deterministic, tree-structured transitions, and binary terminal reward. In this structurally simplified MDP, each action induces a deterministic and new branch, and each partial sequence has exactly one parent state. This critical observation leads us to a central question: whether we are applying unnecessarily complex tools to a structurally simpler (albeit larger) problem, and ***is there a minimalist yet highly effective RLVR algorithm that maintains both quality and diversity under this specialized MDP structure?*** Our theoretical analysis reveals a surprising result under this scenario: the optimal actions can be derived by simply evaluating a fixed uniformly random policy and then selecting actions greedily based on its Q-values. This surprising finding means that we can bypass the standard GPI cycle to identify optimal policies, which requires only policy evaluation of the simplest possible policy (uniformly random), without iterative evaluation of the updated policy and without the many heuristic tricks that plague current methods. Although it was widely believed that this kind of uniform policy is trivial that cannot provide meaningful guidance for control (Asadi & Littman, 2017), the value of uniform policies (He et al., 2025b) has been observed empirically in specific discrete environments (Laidlaw et al., 2023) recently, and we provide a first theoretical analysis account for LLM math reasoning and leverage it as the foundation of our approach.

However, as in standard reward-maximizing RL, while a naive greedy selection guarantees optimality, it sacrifices diversity critical for reasoning tasks (Si et al., 2024). To balance quality and diversity, we leverage a key insight based on our analysis: uniform-policy Q-values capture the probability of successful continuations that lead to positive rewards. As this creates a natural value map of the reasoning landscape, we sample actions via softmax over the uniform-policy Q-values, which maintains performance guarantees while aligning with modern LLM practices (Sheng et al., 2024; Kwon et al., 2023). To translate our theoretical insights into a practical and scalable algorithm for LLM reasoning which involves vast state and action spaces as well as long horizons (a wide and deep tree), we present **R**andom **Po**licy **V**aluation for Di**v**erse **R**easoning (ROVER). ROVER efficiently parameterizes the Q-function intrinsically based on the LLM's parameters, which eliminates the need for a separate value network and also leverages the LLM's strong priors for efficient navigation in the vast token space and stabilizing training through relative improvements. To mitigate the high variance caused by the reward signals, we leverage group reward centering inspired by Naik et al. (2024), and broadcast the reward to improve training efficiency.

Our contributions are as follows: (i) We prove a surprising result: in the deterministic tree-structured MDPs with binary terminal rewards that characterize math reasoning, the optimal action can be derived directly from Q-values evaluated under a uniformly random policy, a finding that fundamentally simplifies RL for this domain. (ii) We introduce ROVER, a practical and minimalist RL

algorithm that is scalable to LLM reasoning tasks through a simplified framework compared to the current complicated methods. (iii) Despite ROVER's radical simplification, extensive experiments across diverse tasks and various model scales demonstrate that it consistently achieves superior performance, yielding **+8.2** improvement on pass@1 and **+16.8** improvement on pass@256 on the competition-level AIME24, AIME25, and HMMT25 tasks. Interestingly, we observe ROVER can find novel reasoning strategies absent from the base model and models trained through standard RL approaches (GRPO), thereby evidencing its potential to push the reasoning boundary. Our codes are available at `https://github.com/tinnerhrhe/ROVER`.

## 2 PRELIMINARIES

**RL with Verifiable Rewards in LLMs.** We investigate reinforcement learning (RL) for post-training LLMs with verifiable rewards, such as mathematical reasoning tasks. We formulate the problem as a Markov Decision Process (MDP), defined by a tuple $(\mathcal{S}, \mathcal{V}, \mathcal{R}, \mathcal{P}, \gamma, \mathcal{X})$. Here, the state space $\mathcal{S}$ denotes all finite-length strings formed by the concatenation of elements in $\mathcal{V}$. The action space $\mathcal{V}$ is the vocabulary set. We set the discount factor $\gamma = 1$ in practice. $\mathcal{R} : \mathcal{S} \times \mathcal{V} \to \mathbb{R}$ is the binary reward function, and $\mathcal{P} : \mathcal{S} \times \mathcal{V} \to \mathcal{S}$ is a deterministic transition function. At the beginning of each episode, a prompt $x$ is sampled from the initial state distribution $\mathcal{X}$. At each step $t$, the LLM selects an action $a_t \in \mathcal{V}$ according to $\pi_\theta(\cdot|s_t)$, and then transits to the next state $s_{t+1} = \{x, a_0, \cdots, a_t\}$ by concatenation. This autoregressive generation continues until forming an entire response $y = \{a_0, a_1, \cdots, a_{|y|-1}\}$, and finally receives a verifiable reward $r(x, y) \in \{0, 1\}$. The goal is to learn a policy $\pi^* = \arg\max_\pi \mathbb{E}_{x \sim \mathcal{X}, y \sim \pi(x)}[r(x, y)]$ by maximizing the expected cumulative reward $r$. The prevailing works leverage policy gradient (Williams, 1992) and a surrogate objective introduced by PPO (Schulman et al., 2017) to optimize $\pi_\theta$:

$$J(\theta) = \mathbb{E}_{x \sim \mathcal{X}, y \sim \pi_{\theta_{\text{old}}}(x)}\left[\frac{1}{|y|}\sum_{t=0}^{|y|-1}\left(\min\left(\text{IS}_t A_t, \text{clip}(\text{IS}_t, 1 - \epsilon_{\text{low}}, 1 + \epsilon_{\text{high}})A_t\right) - \beta D_{KL}(\pi_\theta|\pi_{\text{ref}})\right)\right], \quad (1)$$

where $\text{IS}_t = \pi_\theta(a_t|s_t)/\pi_{\theta_{\text{old}}}(a_t|s_t)$ is the importance sampling ratio, $\pi_{\theta_{\text{old}}}$ is the behavior policy to sample data, $s_t = \{x, a_{<t}\}$ is current state, $\epsilon_{\text{low}}$ and $\epsilon_{\text{high}}$ is the clipping range of importance sampling ratios, $D_{KL}$ denotes the KL regularization term, and $A_t$ is the advantage of current action. $A_t$ is implemented differently across RL algorithms, such as REINFORCE++ (Hu et al., 2025a) and GRPO (Guo et al., 2025). For example, GRPO (Guo et al., 2025) samples $G > 1$ responses for each prompt and estimates the advantage $A_t = \frac{r(x, y_i) - \text{mean}(\{r(x, y_i)\}_{i=1}^G)}{\text{std}(\{r(x, y_i)\}_{i=1}^G)}$ within each group to reduce variance. Notably, while existing policy optimization methods rely on a KL-divergence penalty $(D_{KL})$ to prevent catastrophic forgetting and maintain exploration during continual learning (Liu et al., 2025a), our approach achieves these desiderata without such an explicit regularization term. For a comprehensive discussion of related work, please see Appendix D.

**Generalized Policy Iteration (GPI).** GPI (Sutton et al., 1998) is a unifying view that describes many RL algorithms (e.g., PPO) as illustrated in Fig. 2. GPI consists of two interacting processes, which are *policy evaluation* that estimates how good a policy is, (e.g., via $Q^\pi(s_t, a_t) = r(s_t, a_t) + \gamma\mathbb{E}\left[Q^\pi(s_{t+1}, a_{t+1})\right]$, value function, or advantage function), and *policy improvement* that updates the policy to prefer actions scored better by the current estimates (e.g., $\pi(s) \leftarrow \arg\max_a Q^\pi(s, a)$ or other methods). Littman

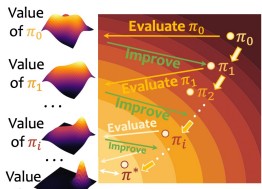

Figure 2: Illustration of GPI.

& Szepesvári (1996) introduced generalized Bellman update which update the Q-function by $\hat{Q}(s_t, a_t) \leftarrow r(s_t, a_t) + \gamma\sum_{s_{t+1} \in \mathcal{S}}\gamma\mathcal{P}(s_t, a_t, s_{t+1})\bigotimes_{a_{t+1}}\hat{Q}(s_{t+1}, a_{t+1})$ with any arbitrary operator $\bigotimes$ that replaces the max operator typically used in Q-learning (Sutton et al., 1998; Haarnoja et al., 2017; Pan et al., 2020), while the mean operator was traditionally considered unsuitable for optimization in general control tasks (Asadi & Littman, 2017). GPI-based methods require an alternative learning over these two processes until finding the fix point, where the learning target remains non-stationary throughout training (Mnih et al., 2015). In contrast, our proposed method relies solely on *policy evaluation* to derive the Q-values of a fixed, uniform random policy, which is much simpler for training and implementation (with a high-level illustration in Fig. 3).

## 3 ROVER: RANDOM POLICY VALUATION FOR DIVERSE REASONING

RLVR for math reasoning can be cast as a decision-making problem in a specialized finite-horizon MDP $\mathcal{M}$ with deterministic transitions and binary terminal rewards (correct or incorrect) in a tree-structured space (each state has a unique parent and actions lead to disjoint subtrees).

This contrasts with general-purpose RL settings that often feature general control problems with stochastic dynamics, complex reward structures, and discrete (or continuous) graph-based state spaces where states can have multiple parents or even cycles. Although the PPO family achieves promising results in LLM reasoning, it was designed for general control

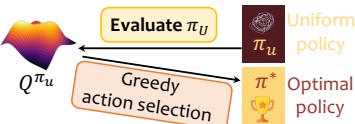

Figure 3: Illustration of ROVER (greedy).

and can encounter entropy and diversity collapse in RLVR, which also introduces unnecessary computational overhead and complexity (Guo et al., 2025).

Motivated by this structural mismatch, we consider an important question overlooked in the literature: *can there exist a minimalist and simple RL approach that exploits these properties of RLVR MDP to achieve both high quality and diversity?* In contrast to adding various implementation-level tricks to PPO/GRPO, we present ROVER, which is built upon a surprising discovery: simply evaluating a uniformly random policy and selecting actions greedily based on its Q-values is sufficient for optimal behavior in this context (Fig. 3), avoiding the complexities of modern deep RL algorithms (Schulman et al., 2017) and can bypass the traditional GPI loop in Fig. 2.

We first establish the theoretical basis of this unexpectedly simple yet optimal approach in § 3.1, extend it to achieve diversity while maintaining performance guarantees in § 3.1.1, and present a practical algorithm that scales to large spaces and long horizons for math reasoning in § 3.2.

### 3.1 THE RANDOM POLICY VALUATION FRAMEWORK

We start from the simplest possible policy, the uniform random policy $\pi_u(a|s) = \frac{1}{|A|}$, where $A$ denotes the set of available actions. The corresponding Q-value for $\pi_u$ can be estimated using the generalized Bellman update (Littman & Szepesvári, 1996; Sutton et al., 1998) with the mean operator (Asadi & Littman, 2017). The mean operator corresponds to evaluating a uniform policy, and the update is simplified to $\hat{Q}^{\pi_u}(s,a) \leftarrow r(s,a) + \frac{1}{|A|} \sum_{a' \in \mathcal{A}} \hat{Q}^{\pi_u}(s',a')$ for deterministic transitions and $\gamma = 1$ (Hu et al., 2025b) that we consider as discussed in § 2. The literature of classical RL suggests that this mean operator is insufficient for optimal control in general MDPs (Asadi & Littman, 2017), as it averages across all actions without preference for optimal ones, providing little guidance. While a few recent studies have empirically noted the potential utility of uniform-policy values in certain discrete games (Laidlaw et al., 2023; He et al., 2025b), these observations have remained primarily empirical, with limited theoretical justification.

In our context, LLM math reasoning induces finite-horizon, deterministic, tree-structured MDPs with binary terminal rewards (correct/incorrect). For a root state $s_0 = x$ (i.e., prompt), the reachable transition graph is a rooted tree, where each state has a unique path from $s_0$ and distinct actions from a state lead to disjoint subtrees. Under this context, we prove that simply evaluating the fixed uniform policy and acting greedily with respect to its Q-values already achieves optimality in Theorem 1. The proof can be found in Appendix A.1.

**Theorem 1.** *Consider a finite-horizon episodic MDP with deterministic transitions, tree-structured state space, and binary terminal rewards $\mathcal{R}(s) \in \{0, R\}$ where $R > 0$ ($R$ for a correct solution, $0$ otherwise). Let $\pi_u$ be the uniform policy, and $Q^{\pi_u}$ its corresponding Q-function. Define the greedy policy with respect to $Q^{\pi_u}$ by $\pi_{\text{greedy}}(s) = \arg\max_a Q^{\pi_u}(s,a)$, then $\pi_{\text{greedy}}$ is optimal.*

From Theorem 1, we discover that for the specific MDP structure of LLM math reasoning, the optimal control problem reduces to a much simpler form than previously recognized. This suggests two significant implications: First, despite the perceived complexity of LLM math reasoning tasks, their underlying decision structure exhibits a more tractable structure than commonly assumed. Second, the mean operator, although generally insufficient for optimal control, proves to be surprisingly powerful when paired with a greedy action selection strategy in this context.

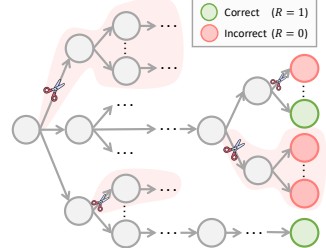

Figure 4: Intuition of ROVER (greedy) with $\pi_{\text{greedy}}$.

Surprisingly, although the uniformly random policy itself is far from optimal behavior, its Q-values have a meaningful interpretation here, which equals the probability that, after taking $a$ at $s$ and then acting uniformly at random until termination, we obtain a correct outcome. As illustrated in Fig. 4, when $Q^{\pi_u}(s,a) = 0$, it indicates that no possible continuation from $(s,a)$ can lead to a correct solution. Conversely,

higher values indicate more promising directions. By acting greedily with respect to these values, we effectively eliminate branches that cannot lead to valid solutions while prioritizing the most promising paths. This property enables optimality through a remarkably computationally simple mechanism: we need only estimate $Q^{\pi_u}(s, a)$ by policy evaluation for a fixed uniform policy $\pi_u$, without off-policy corrections or the implementation complexity of popular methods like PPO and GRPO. Additionally, since our approach evaluates a fixed uniform policy rather than iteratively improving a learned policy, it mitigates the non-stationarity issues that plague many modern deep RL methods (Van Hasselt et al., 2016), which can also be advantageous for the high-dimensional, complex LLM math reasoning tasks.

**A Didactic Example.** To empirically validate the optimality of the greedy policy derived from the Q-function of a uniformly random policy, we design a tabular environment as illustrated in Fig. 5(a). The environment is a deterministic, tree-structured MDP capturing the essential properties of LLM math reasoning tasks while remaining transparent for analysis (and we will introduce how to scale up the method in § 3.2). Starting from an initial null state, a policy executes an action $a \sim \mathcal{A} = \{A, B, C, D\}$ by appending it to the current state sequence. We consider an episodic setup with binary terminal rewards, with 4 specific terminal states (ACD, BDC, CAB, DBA) yielding a reward of 1 and all others yielding 0. From Fig. 5(c), we observe that the simple mechanism of acting greedily with respect to a random policy's Q-function also learns to generate the sequence with the highest reward, achieving the same optimal behavior as Q-learning (with $\epsilon$-greedy exploration).

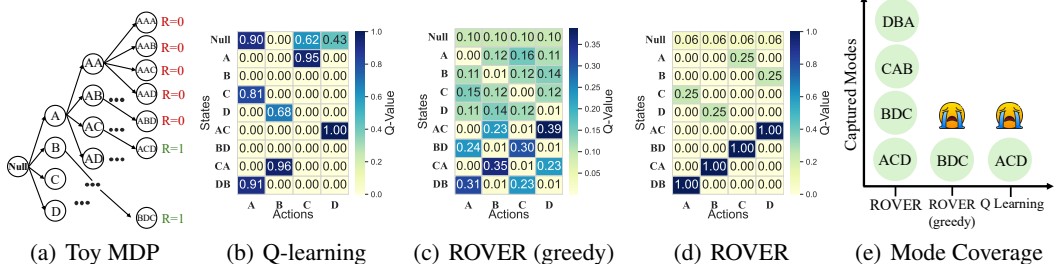

|   (a) Toy MDP   |   (b) Q-learning   |   (c) ROVER (greedy)   |   (d) ROVER   |   (e) Mode Coverage   |

Figure 5: (a) Illustration of the tabular MDP. (b)-(d) Comparison of learned Q-value maps. According to the Q-values, standard Q-learning with $\epsilon$-greedy exploration converges to the mode ACD. ROVER (greedy) assigns the highest Q-values to optimal actions, but still converges to a single mode BDC due to its greedy behavior. ROVER is able to assign equally high Q-values to all optimal actions. (e) Q-learning and ROVER (greedy) converge to a single mode despite both being optimal, whereas ROVER successfully covers all 4 optimal modes.

### 3.1.1 BEYOND GREEDY SELECTION: BALANCING QUALITY AND DIVERSITY

While our theoretical analysis shows that the simple scheme of greedy selection over the Q-values of a uniform policy is already enough for achieving optimality, this deterministic approach often leads to mode collapse and sacrifices diversity (Fig. 5(e)). For LLM math reasoning tasks, as a given prompt can elicit multiple viable responses that yield correct solutions, diversity is critical for robust problem-solving (Li et al., 2025a), which is also important for improving pass@$k$ performance and generalization to novel problems.

Our analysis reveals a key insight: the $Q^{\pi_u}(s, a)$ characterizes the probability of successful continuations following the action $a$, where higher Q-values indicate action branches with denser successful pathways. To improve the diversity of policy generation, based on this insight, we transition from deterministic to stochastic action selection by converting $Q^{\pi_u}$ into a soft sampler, i.e., $\pi_s(a|s) = \frac{\exp(Q^{\pi_u}(s,a)/\rho)}{\sum_{a'} \exp(Q^{\pi_u}(s,a')/\rho)}$, where $\rho$ is a temperature parameter. This strategy selects actions proportional to their estimated success probability, which is able to explore multiple reasoning pathways for improving diversity, rather than committing to a single path. Additionally, it aligns with contemporary LLM decoding strategies (Kwon et al., 2023), making it readily integrable into existing training frameworks (Sheng et al., 2024). The following result shows that our softmaxing $Q^{\pi_u}$ approach maintains a guaranteed level of performance relative to the optimal policy, with the bound tightening as temperature decreases. The proof can be found in Appendix A.2.

**Theorem 2.** *Consider the same MDP $\mathcal{M}$, and let $Q^{\pi_u}(s, a)$ denote the Q-function under the uniform random policy $\pi_u$ from state-action pair $(s, a)$, $N(s) = |\{a : Q^{\pi_u}(s, a) = 0\}|$ be the number of*

*zero-valued actions at state $s$, $A(s)$ be the number of available actions at state $s$, and $P$ denotes the set of key states where both optimal and suboptimal actions exist, i.e., $P = \{s : 1 \leq N(s) \leq A(s) - 1\}$. Given the softmax policy $\pi_s(a|s) = \frac{\exp(Q^{\pi_u}(s,a)/\rho)}{\sum_{a'} \exp(Q^{\pi_u}(s,a')/\rho)}$ with temperature $\rho > 0$, and $Pr^{\pi_s}(s|s_0)$ is the probability of reaching $s$ from $s_0$ with the policy $\pi_s$, the value function of the induced policy $\pi_s$ satisfies: $V^{\pi_s}(s_0) \geq R \left( 1 - \sum_{s \in P} Pr^{\pi_s}(s|s_0) \frac{N(s)}{N(s) + \exp(\max_a Q^{\pi_u}(s,a)/\rho)} \right)$.*

Theorem 2 characterizes that the temperature $\rho$ trades off between diversity and quality. As $\rho$ increases, the policy samples more diverse actions while still favoring higher-value paths. When $\rho$ approaches zero, the performance gap between the softmax policy and the optimal policy vanishes, showing that our diversity-promoting approach maintains performance guarantees.

**Justification.** In our didactic example (Fig. 5(d)&5(e)), we empirically demonstrate that it achieves an effective tradeoff. While both greedy approaches (Q-learning and ROVER (greedy)) achieve optimal reward but collapse to a single solution mode, ROVER (with $\rho = 1$) successfully identifies all four optimal modes while maintaining 100% success rate. Our diversity-seeking RL approach stands in contrast to typical RL diversity methods that often rely on complex and task-related reward engineering (He et al., 2025a; Cheng et al., 2025; Li et al., 2025a) or post-hoc sampling techniques (Shur-Ofry et al., 2024; Chen et al., 2025b) without guarantees, while remaining simple.

## 3.2 PRACTICAL IMPLEMENTATION

We now adapt our method to LLMs, where the induced MDP still remains deterministic and tree-structured, but presents computational challenges due to long horizons (deep trees) and large vocabularies (wide branching). To address these challenges, we introduce practical techniques to approximate, stabilize the training process, and improve sample efficiency as summarized in Alg. 1, while preserving the core idea of random policy evaluation. We also provide gradient analysis and connections to policy gradient methods in Appendix B.

---

**Algorithm 1:** Random Policy Valuation for Diverse Reasoning (ROVER)

**Input:** pre-trained LLM $\pi_\theta$, epochs M, prompt dataset $\mathcal{D}$, group size $n$, lr $\eta$, temperature $\rho$

1 **for** *epoch $m = \{1, \cdots, M\}$* **do**
2      Set $\pi_{\theta_{\text{old}}} \leftarrow \pi_\theta$; Sample a batch of prompts $\mathcal{B} \sim \mathcal{D}$ via $\pi_{\theta_{\text{old}}}$
3      **for** *each prompt $x \in \mathcal{B}$* **do**
4          Rollout responses and compute rewards: $\{y_i\}_{i=1}^n \sim \pi_{\theta_{\text{old}}}(\cdot|x)$; $\tilde{r} = r_i - \frac{1}{n} \sum_{i=1}^n r_i$
5      **for** *each prompt-response pair $\{x, y_i\}$ in batch* **do**
6          **for** *each state $s_t \in \{x, y_i\}$* **do**
7              Compute Q-value $Q(a_{t+1}|s_{t+1}) = \rho\big( \log \pi_\theta(a_{t+1}|s_{t+1}) - \log \pi_{\theta_{\text{old}}}(a_{t+1}|s_{t+1}) \big)$
8              Obtain $\hat{Q}(a_t|s_t) \leftarrow \tilde{r} + \frac{1}{|\mathcal{V}|} \sum_{a_{t+1} \in \mathcal{V}} Q(a_{t+1}|s_{t+1})$ // $\mathcal{V}$: the vocabulary set.
9          $\mathcal{L}_{\text{ROVER}} = \frac{1}{\sum_{i=1}^n |y_i|} \sum_{i=1}^n \sum_{t=0}^{|y_i|-1} \|Q(a_t|s_t), \text{sg}[\hat{Q}(a_t|s_t)]\|^2$ // sg: stop gradient.
10          $\theta \leftarrow \theta - \eta \nabla_\theta \mathcal{L}_{\text{ROVER}}$ by an AdamW optimizer

---

• **Q Parameterization.** While we begin with a reference LLM, we lack a pre-trained Q-function. Training a Q-model from scratch presents substantial costs due to the large scale of action and state spaces. A compelling approach is to represent the Q-function directly through the LLM's intrinsic parameters $\theta$ (Li et al., 2025b), thereby eliminating the need for a separate value network. Fortunately, as indicated in Theorem 2 and following the mean operator for evaluating the value of the uniform policy in § 3.1, the policy $\pi_\theta$ and Q-values can be intrinsically linked through $\rho \log \pi_\theta(a_t|s_t)$ with $\rho$ the temperature, which captures the relative preference over actions within each state (though a state-dependent constant term is omitted for efficient practical implementation). However, this direct formulation is unstable in practice since the learning target drifts as the policy changes and the Q-value updates are prone to divergence. To mitigate this instability, we introduce a relative Q-function that measures the improvement over a fixed baseline: $Q(s_t, a_t) = \rho\big( \log \pi_\theta(a_t|s_t) - \log \pi_{\theta_{\text{old}}}(a_t|s_t) \big)$, where $\pi_{\theta_{\text{old}}}$ is the behavior policy used to sample data in each epoch, serving as a stable anchor that reduces fluctuations. This parameterization centers the initial Q-values around zero and ensures the model learns the change relative to the previous policy instead of absolute values.

• **Low-Variance Reward.** To create a stable and dense reward signal for learning uniform-policy Q-values, we sample $n$ responses for each prompt to reduce estimation variance and enrich our approximation of the value landscape. Inspired by Naik et al. (2024), we subtract the empirical average reward of the $n$ responses from the raw rewards to obtain mean-centered rewards. Specifically, the centered reward is given by $\tilde{r}(x, y_i) = r(x, y_i) - \frac{1}{n}\sum_{i=1}^{n} r(x, y_i)$, where $r(x, y_i)$ reflects the correctness of the corresponding response $y_i$ given the prompt $x$. This is also related to GRPO's style of estimating the advantage function, but without the standard deviation normalization term (Liu et al., 2025c). Additionally, to ensure efficient credit assignment, especially for long reasoning chains, we broadcast this centered reward $\tilde{r}(x, y_i)$ to every token in the generation following Hu et al. (2025b) .

## 4 EXPERIMENTS

Although simple, our method substantially enhances both the quality and diversity of LLM generations, leading to improved reasoning capabilities on complex tasks. We evaluate our approach on two verifiable tasks that require sophisticated reasoning: countdown tasks, which have multiple valid answers, and math competitions, which possess single, unambiguous answers.

### 4.1 COUNTDOWN TASKS

We begin evaluating our method on the countdown task. Given an array of numbers and a target, the LLM must find the correct sequence using the four basic arithmetic operations $(+, -, \times, \div)$ to reach the target number. We selected Countdown since it offers a restricted search space and multiple valid answers for a question that enables tractable analysis of both the reasoning behavior and diversity.

**Setup.** We evaluate on the TinyZero (Pan et al., 2025) dataset with 1,024 test problems. We employ Qwen2.5-3B (Team, 2024) as our base model, which demonstrates near-zero accuracy on this specific task that establishes a clear baseline for improvement. We benchmark our method against the well-recognized GRPO (Shao et al., 2024) and two GRPO variants designed for policy entropy preservation: one with varying KL coefficients and another incorporating the clip-higher technique (Yu et al., 2025). Detailed task descriptions and the training details are in Appendix E.

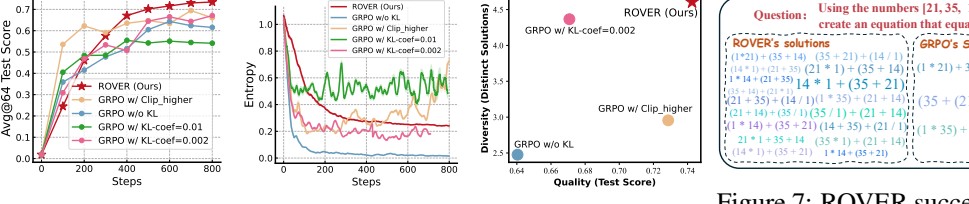

(a) Test Score     (b) Entropy     (c) Diversity&Quality

Figure 6: Performance of our method and baselines over training on countdown tasks. The y-axis of (c) denotes the number of found distinct correct solution equations, averaged over 1024 questions.

Figure 7: ROVER successfully finds 17 diverse solution equations, while only 3 different equations are given by GRPO.

**Results Analysis.** From the results shown in Fig. 6, we have the following observations: (i) In terms of test scores shown in Fig. 6(a), our method surpasses all baselines after 400 training steps, ultimately reaching the highest ceiling performance. Conversely, the GRPO with a KL coefficient of 0.01 performs distinctly worse, indicating that its performance is hampered by excessive regularization. We attribute the efficacy of our method to the preservation of high policy entropy throughout training. As shown in Fig. 6(b), our method's entropy decays gracefully while remaining significantly higher than that of the baselines, which either collapse (GRPO w/o KL) or fluctuate erratically (GRPO w/ Clip_higher). A stable high entropy encourages sustained exploration, which is the primary driver of our model's performance, enabling it to achieve the highest scores on both quality and diversity metrics, as validated in Fig. 6(c), where our method finds more diverse solutions to address a question. Fig. 7 further provides a visualization example to demonstrate the solution diversity of ROVER .

**Ablation on temperature $\rho$.** Consistent with standard LLM sampling practices (Sheng et al., 2024), we set temperature $\rho = 1$ for softmax sampling for all experiments without any task-specific tuning. This parameter balances the exploration-exploitation trade-off: $\rho \rightarrow 0$ encourages greedy, deterministic behavior, while higher values promote diverse sampling.

Our ablation study on $\rho$ in Fig. 8 confirms that $\rho = 1$ achieves a robust and desirable performance. A higher temperature causes under-exploitation and slower convergence, while a lower value triggers premature exploitation, causing an accelerated collapse in policy entropy and constrained exploration space. In the extreme case where $\rho = 0.001$, the near-deterministic policy sampling leads to severe training instability (evidenced in test score), highlighting the importance of a balanced temperature for effective exploration. We

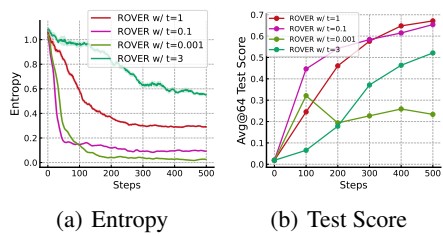

(a) Entropy      (b) Test Score

Figure 8: Performance under different $\rho$.

further investigate the effect of $\rho$ on math reasoning tasks, where similar conclusions are validated. Results are provided in Appendix G.3.1.

## 4.2 REASONING ON MATH TASKS

**Training Setup.** We employ models of various sizes for validating the efficacy of our proposed method, including Qwen3-8B-Base, Qwen3-4B-Base, and DeepSeek-R1-Distill-Qwen-1.5B, where the results of DeepSeek-1.5B are provided and analyzed in Appendix F due to space limitations. All models are trained on the open-source DeepScaler dataset (Luo et al., 2025). A binary reward is assigned by the open-source verification tool `math_verify` (Kydlíček & Face, 2025) upon the completion of LLM generation. We employ standard RLVR methods as baselines, including GRPO (Shao et al., 2024), REINFORCE++ (Hu et al., 2025a), and DAPO (Yu et al., 2025).

**Evaluation.** We select various widely-acknowledged math reasoning benchmarks: AIME24 (MAA, 2024), AIME25 (MAA, 2025), HMMT25 (Balunović et al., 2025), OlympiadBench (He et al., 2024), AMC23 (AI-MO, 2024), and MATH500 (Hendrycks et al., 2021), along with the O.O.D benchmark GPQA-diamond (Rein et al., 2024). We report pass@1 and pass@$k$ for comprehensive analysis, where pass@$k$ measures diversity and the reasoning boundary (Yue et al., 2025). With increased diversity, the model has a higher probability of discovering a correct reasoning path within $k$ attempts. More details about the experimental setup can be found in the Appendix G.1.

Table 1: Pass@1 results across different methods on mathematical and O.O.D benchmarks. The highest and the second-best scores are shown in bold and underlined, respectively.

| Pass@1 | Mathematical | | | | | | O.O.D | Avg. |
|---|---|---|---|---|---|---|---|---|
| | AIME 2024 | AIME 2025 | HMMT 2025 | Olympiad Bench | AMC 2023 | MATH 500 | GPQA diamond | |
| *Qwen3-4B-Base* | | | | | | | | |
| Base Model | 8.8 | 4.9 | 0.8 | 27.3 | 35.2 | 55.6 | 9.7 | 20.3 |
| GRPO | 16.4 | 9.4 | 2.4 | 43.6 | 57.0 | 79.9 | 38.7 | 35.3 |
| DAPO | 17.1 | 10.9 | 0.7 | 41.7 | 56.6 | 78.4 | 38.5 | 34.8 |
| REINFORCE++ | 14.8 | 7.8 | 2.8 | 42.3 | 57.9 | 76.8 | 31.8 | 33.5 |
| **ROVER (Ours)** | **17.6** ↑ +8.8 | **12.6** ↑ +7.7 | **3.1** ↑ +2.3 | **45.4** ↑ +18.1 | 57.1 ↑ +21.9 | **80.5** ↑ +24.9 | **39.5** ↑ +29.8 | **36.5** ↑ +16.2 |
| *Qwen3-8B-Base* | | | | | | | | |
| Base Model | 11.5 | 8.8 | 0.8 | 34.7 | 48.1 | 68.8 | 29.1 | 28.8 |
| GRPO | 16.8 | 15.1 | 4.8 | 48.6 | 66.9 | 81.9 | 43.8 | 39.7 |
| DAPO | 20.8 | 15.2 | 3.6 | 49.0 | 67.9 | 84.3 | 46.6 | 41.1 |
| REINFORCE++ | 19.4 | 16.7 | 7.1 | 47.6 | 63.5 | 83.6 | 46.3 | 40.6 |
| **ROVER (Ours)** | **30.6** ↑ +19.1 | **22.7** ↑ +13.9 | **14.6** ↑ +13.8 | **56.4** ↑ +21.7 | **74.8** ↑ +26.7 | **89.6** ↑ +20.8 | **50.2** ↑ +21.1 | **48.4** ↑ +19.6 |

### 4.2.1 PERFORMANCE ANALYSIS

**ROVER consistently outperforms all RL baselines in terms of average pass@1.** As detailed in Table 1, ROVER consistently outperforms standard RL methods across all model sizes. For the Qwen3-8B-Base model, ROVER achieves pass@1 improvements of **+7.3** and **+8.2** over the strongest baseline, averaged on all benchmarks and on the subset of AIME24, AIME25 and HMMT25, respectively. The superiority of our method over baseline methods becomes more pronounced on increasingly challenging tasks. Notably, for Qwen3-8B-Base, ROVER delivers substantial relative improvements of **+47.1%** on AIME24 and **+35.9%** on AIME25 over the best-performing baseline. On HMMT25, ROVER nearly doubles the performance of the strongest baseline, REINFORCE++.

**ROVER significantly improves pass@$k$.** The average pass@$k$ over a dataset reflects the proportion of problems a model can potentially solve within $k$ trials, serving as a robust evaluation metric of the model's reasoning breadth and diversity. To demonstrate the effectiveness of our method in incentivizing reasoning diversity, we compare ROVER with baselines by scaling pass@$k$ from 1

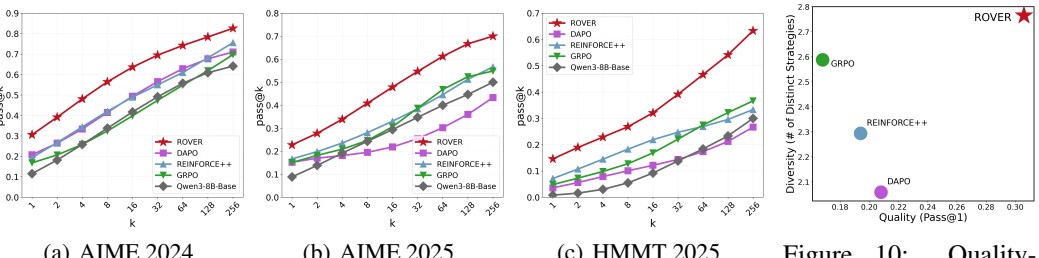

(a) AIME 2024  (b) AIME 2025  (c) HMMT 2025

Figure 9: pass@$k$ of ROVER and baselines on Qwen3-8B-Base.

Figure 10: Quality-Diversity tradeoff.

to 256. Consistent with previous observations (Yue et al., 2025; Li et al., 2025a), the results in Fig. 9 reveal that while standard RL baseline methods enhance pass@1, their performance quickly saturates and plateaus, ultimately underperforming the base model at large $k$ values. For example, DAPO even shows worse performance on AIME25 after $k > 4$, a trend that is also observed on HMMT25 for $k > 32$. In contrast, our method demonstrates sustained and significant performance gains as $k$ increases, consistently surpassing all the baselines and the base model (**+16.8** over the best baseline on pass@256 averaged on AIME24, AIME25 and HMMT25). This advantage is particularly pronounced on the most challenging HMMT25 task, where our method's pass@$k$ score continues to accelerate while all baselines have saturated. We attribute the improved pass@$k$ to ROVER's ability to maintain a relatively higher entropy during training (see Fig. 22), which ensures sustained exploration of different reasoning strategies and enhances reasoning diversity.

**ROVER shows remarkable generalization on O.O.D tasks**. To further evaluate the generalization capability of ROVER, we incorporate the GPQA-diamond benchmark, a challenging math-unrelated task containing 198 graduate-level questions in biology, physics, and chemistry. The results in Table 1 demonstrate ROVER's stronger generalization beyond the training distribution, achieving the best performance on the unseen GPQA-diamond benchmark.

### 4.2.2 DIVERSITY ANALYSIS

**ROVER possesses the highest diversity across different metrics**. To quantify reasoning diversity, we employ the "number of distinct strategies" metric from NoveltyBench (Zhang et al., 2025c). Specifically, we sample up to 32 correct responses for each problem from the AIME24 datasets, and leverage Claude-3.5-Sonnet as the LLM judger to determine strategic equivalence between these response pairs (template in Fig. 27). A higher number of distinct strategies (classes) indicates greater reasoning diversity. We report the results in Fig. 10 (with a 0.6 decoding temperature) and the results across different decoding temperatures in Fig. 24. From Fig. 10, we observe that ROVER demonstrates relative diversity improvements of **+6.8%** and **+20.5%** when compared with GRPO and the average of all three baselines, respectively. Conventional RL approaches struggle to improve diversity merely through increasing sampling temperature during inference, while ROVER consistently improves the Pareto front between quality and diversity. For a more comprehensive quantitative analysis of generation diversity, we refer to Appendix G.4, which includes results for additional metrics such as utility (Zhang et al., 2025c) and cosine distance (Fig. 25).

### 4.2.3 BEHAVIORAL ANALYSIS

**ROVER scales best at test-time due to maintained diversity**. Test-time scaling has received significant attention due to its potential to enhance reasoning performance, where majority voting is a fundamental baseline for evaluating LLM scalability at test-time (Liu et al., 2025b). Fig. 11 confirms that ROVER's maj@$k$ performance scales robustly, consistently improving upon the base model across all $k$ values, even on the most challenging HMMT25 task. This superior scalability stems from ROVER's ability to maintain a diverse distribution over valid

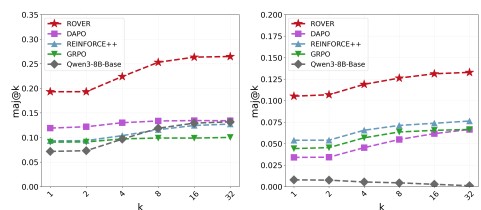

(a) AIME 2024  (b) HMMT 2025

Figure 11: Maj@$k$ performance of ROVER and baselines on Qwen3-8B-Base.

reasoning paths, while baseline methods suffer from mode collapse, causing them to confidently converge on similar incorrect solutions and preventing performance gains from additional samples.

**Enhanced reflection behaviors**. To analyze the reasoning patterns learned via ROVER, we adopt the *forking tokens* defined in Wang et al. (2025) (see Table 8) and quantify the normalized frequency of these tokens in the generated outputs (256 rollouts per prompt on AIME24, AIME25, and HMMT25). Fig. 12 shows models trained with ROVER generate a significantly higher proportion of these *forking tokens*, particularly those associated with rethinking and self-correction (e.g., 'wait' and 'however'). As detailed in Fig. 18, ROVER encourages the model to actively reflect upon, verify, and pivot

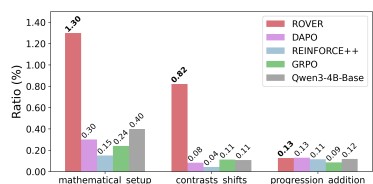

Figure 12: Comparison of reflection frequency. ROVER outputs more reasoning-related tokens.

between different reasoning strategies, rather than committing to a single reasoning path. Fig. 17 examines an AIME24 problem where ROVER discovers two additional novel strategies compared to the base and GRPO-trained models, showing ROVER's potential to push the reasoning boundary.

## 4.3 Reasoning on Tool-Use Tasks

We further validate that ROVER performs well in tool-use scenarios. Specifically, following the Search-R1 Jin et al. (2025) setup, we fine-tune the Qwen2.5-3B-Base model on the combined training sets of Natural Questions (NQ) and HotpotQA. We evaluate ROVER after 300 training steps. Table 11 presents the performance across multiple benchmarks, spanning both General QA and Multi-hop QA tasks. The results show that ROVER not only works effectively with tools but also outperforms the original GRPO in terms of average performance by 1.8%.

Table 2: Performance comparison on General QA and Multi-hop QA benchmarks.

| | General QA | | | Multi-hop QA | | | Avg. |
|---|---|---|---|---|---|---|---|
| | NQ | TriviaQA | PopQA | HotpotQA | 2wiki | Musique | |
| Search-R1-Base (GRPO) (Jin et al., 2025) | 0.421 | 0.583 | 0.413 | 0.297 | 0.274 | 0.066 | 0.342 |
| ROVER | **0.477** | **0.600** | **0.443** | 0.290 | **0.286** | **0.069** | **0.360** |

**Diversity comparison.** To evaluate diversity, we generate 32 responses for each prompt from the Bamboogle benchmark with temperature set to 1.0, using the cosine distance between agent responses as the diversity metric. As shown in Table 12, ROVER demonstrates substantially better diversity. An agent capable of generating diverse queries can retrieve more varied and comprehensive information from the corpus, thereby increasing its chances of discovering the key evidence required to answer a prompt correctly. Conversely, a model lacking diversity may fall into monotonous query patterns, leading to a failure loop where it repeatedly fetches the same unhelpful results from the search engine and fails to solve the problem.

Table 3: Diversity comparison evaluated on Bamboogle benchmark using cosine distance between agent responses.

| Method | Cosine Distance |
|---|---|
| Search-R1-Base (GRPO) (Jin et al., 2025) | 0.042 |
| ROVER | **0.065** |

## 5 Conclusion and Limitation

We present ROVER, a minimalist approach to RLVR that achieves high-quality and diverse reasoning policies from uniformly random valuation, which eliminates the need for complex evaluation-improvement loops with superior performance and diversity compared to existing SOTA methods.

Our experiments are limited to math reasoning tasks with models up to 8B parameters due to restricted computational resources. The practical implementation of ROVER for scaling up to large action spaces and long horizons also introduces approximation. Although the empirical success suggests robustness in the underlying principles despite these approximations, an interesting future direction is to further bridge this gap. We consider these as opportunities to reconsider RLVR from the first principles, develop more robust simplified approaches, and extend ROVER to other tasks (from capable pre-trained models). We believe that our approach establishes a valuable foundation for future research by demonstrating the power of a surprising simplification in this domain, and hope that it inspires future research to adapt and extend these insights to other structures while maintaining the core benefits of simplicity for high-quality performance and diversity preservation.

## ACKNOWLEDGMENTS

This work of Haoran He, Yuxiao Ye, and Ling Pan is supported by National Natural Science Foundation of China 62406266. This work was supported by computing resources and infrastructure provided by StepFun. We are grateful to researchers from StepFun for their valuable feedback and contributions.

## REPRODUCIBILITY STATEMENT

To ensure reproducibility, we provide detailed proofs of our theoretical results in Appendix A. Detailed experimental setup and hyperparameters used during training and evaluation can be found in Appendix G.1. Moreover, we provide the anonymous codebase at `https://anonymous.4open.science/r/ROVER`.

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

## A    PROOFS IN § 3.1

### A.1    PROOF OF THEOREM 1

**Theorem 1** *Consider an episodic finite-horizon episodic MDP with binary terminal rewards $\mathcal{R}(s) \in \{0, R\}$ where $R > 0$ ($R$ for a correct solution, $0$ otherwise). Let $\pi_u$ be a uniform policy, and let $Q^{\pi_u}$ denote its Q-function. Define the greedy policy with respect to $Q^{\pi_u}$ by $\pi_{\text{greedy}}(s) = \arg\max_a Q^{\pi_u}(s, a)$. Then $\pi_{\text{greedy}}$ is the optimal policy.*

*Proof.* As the underlying graph is a tree, starting from $s_0$ under policy $\pi_{\text{greedy}}$ gives a unique chain $s_0 \to s_1 \to \cdots \to s_n$. By definition, for any state-action pair $(s, a)$, if the subtree below $(s, a)$ does not contain a correct terminal state, then $Q^{\pi_u}(s, a) = 0$; conversely, if its subtree contains a correct terminal state, then $Q^{\pi_u}(s, a) > 0$. Therefore, at $s_0$ we choose $a_0 = \arg\max_a Q^{\pi_u}(s, a)$, the next state $s_1$ will necessarily lie on a path that reaches a correct terminal state. We keep proceeding until $s_{n-1}$, and $\pi_{\text{greedy}}(a|s_{n-1}) = \arg\max_a Q^{\pi_u}(s_{n-1}, a)$ also selects the optimal action $a$ (as $Q^{\pi_u}(s_{n-1}, a_{n-1}) = R(s_{n-1}, a_{n-1}) = R$).

$\square$

### A.2    PROOF OF THEOREM 2

**Theorem 2** *Consider the same MDP $\mathcal{M}$, and let $Q^{\pi_u}(s, a)$ denote the Q-function under the uniform random policy $\pi_u$ from state-action pair $(s, a)$, $N(s) = |\{a : Q^{\pi_u}(s, a) = 0\}|$ be the number of zero-valued actions at state $s$, $A(s)$ be the number of available actions at state $s$, and $P$ denotes the set of key states where both optimal and suboptimal actions exist, i.e., $P = \{s : 1 \leq N(s) \leq A(s) - 1\}$. Given the softmax policy $\pi_s(a|s) = \frac{\exp(Q^{\pi_u}(s,a)/\rho)}{\sum_{a'} \exp(Q^{\pi_u}(s,a')/\rho)}$ with temperature $\rho > 0$, and $Pr^{\pi_s}(s|s_0)$ is the probability of reaching $s$ from $s_0$ with the policy $\pi_s$, the value function of the induced policy $\pi_s$ satisfies the following lower bound: $V^{\pi_s}(s_0) \geq R\left(1 - \sum_{s \in P} Pr^{\pi_s}(s|s_0) \frac{N(s)}{N(s) + \exp(\max_a Q^{\pi_u}(s,a)/\rho)}\right)$.*

*Proof.* Let us sample trajectories from the initial state $s_0$ using policy $\pi_s$. For any incorrect trajectory $\tau$ that achieves a reward value of $0$ (one that fails to reach a correct terminal state with positive reward $R$), there must exist at least one key state along $\tau$. For each $\tau$, let $s_\tau$ denote the last key state along $\tau$.

The probability of trajectory $\tau$ can be factored as:

$$Pr(\tau) = Pr^{\pi_s}(s_\tau|s_0) \prod_{t \geq t_\tau} \pi_s(a_t|s_t), \tag{2}$$

where $t_\tau$ denotes the index of state $s_\tau$ in the trajectory sequence.

Let $\mathcal{T}_w$ denote the set of all incorrect trajectories, then we have that

$$Pr(\mathcal{T}_w) = \sum_{\tau \in \mathcal{T}_w} Pr(\tau). \tag{3}$$

For any key state $s \in P$, let $\mathcal{T}(s)$ denote the set of incorrect trajectories for which $s$ is the last key state. Since the underlying MDP $\mathcal{M}$ has a tree structure, the sets $\{\mathcal{T}(s)\}_{s \in P}$ form a partition of $\mathcal{T}_w$. Therefore, we have that

$$Pr(\mathcal{T}_w) = \sum_{s \in P} Pr^{\pi_s}(s|s_0) \sum_{\tau \in \mathcal{T}(s)} \prod_{t \geq t_s} \pi_s(a_t|s_t). \tag{4}$$

As the state $s$ is the last key state on any trajectory in $\mathcal{T}(s)$, we have that

$$\sum_{\tau \in \mathcal{T}(s)} \prod_{t \geq t_s} \pi_s(a_t|s_t) = Pr(Q^{\pi_u}(s,a) = 0|s, \pi_s), \tag{5}$$

where $Pr(Q^{\pi_u}(s,a) = 0|s, \pi_s)$ is the probability that policy $\pi_s$ selects an action with zero Q-value at state $s$.

By the definition of the softmax policy, we have that

$$Pr(Q^{\pi_u}(s,a) = 0|s, \pi_s) = \sum_{a: Q^{\pi_u}(s,a)=0} \pi_s(a|s) \tag{6}$$

$$= \sum_{a: Q^{\pi_u}(s,a)=0} \frac{\exp(Q^{\pi_u}(s,a)/\rho)}{\sum_{a'} \exp(Q^{\pi_u}(s,a')/\rho)} \tag{7}$$

$$= \frac{N(s)}{N(s) + \sum_{a': Q^{\pi_u}(s,a')>0} \exp(Q^{\pi_u}(s,a')/\rho)} \tag{8}$$

$$\leq \frac{N(s)}{N(s) + \exp(\max_{a'} Q^{\pi_u}(s,a')/\rho)} \tag{9}$$

Combing Eq. (4), Eq. (5), and Eq. (9), we have that

$$Pr(\mathcal{T}_w) \leq \sum_{s \in P} Pr^{\pi_s}(s|s_0) \frac{N(s)}{N(s) + \exp(\max_{a'} Q^{\pi_u}(s,a')/\rho)}. \tag{10}$$

By definition, the value function of $\pi_s$ is related to the probability of correct trajectories:

$$V^{\pi_s}(s_0) = (1 - Pr(\mathcal{T}_w))R. \tag{11}$$

Substituting our upper bound on $Pr(\mathcal{T}_w)$, we have that

$$V^{\pi_s}(s_0) \geq R\left(1 - \sum_{s \in P} Pr^{\pi_s}(s|s_0) \frac{N(s)}{N(s) + \exp(\max_{a'} Q^{\pi_u}(s,a')/\rho)}\right). \tag{12}$$

For any key state $s \in P$, we have that $\max_{a'} Q^{\pi_u}(s,a') > 0$ by definition. As $\rho \to 0$, the right-hand side in Eq. (12) converges to $R$, which is the optimal value. $\square$

## A.3 Additional Theoretical Analysis

**Theorem 3.** *Consider a finite-horizon MDP with deterministic transitions and tree-structured state space. Let $\pi_u$ be a uniform policy, $Q^{\pi_u}$ denote its Q-function, and the discount factor $\gamma = 1$. For any non-terminal state $s$, define $\mathcal{C}$ to be the set of terminal states with correct rewards, $T(s,a)$ the next state reached by taking action $a$ from state $s$, and $\mathcal{T}(s)$ the set of all possible trajectories from state $s$ to terminal states. Define two disjoint subsets of actions at state $s$, where $\mathcal{A}^+(s) = \{a \in \mathcal{A}|\exists \tau \in \mathcal{T}(T(s,a))$ such that $\tau$ ends in $\mathcal{C}\}$, and $\mathcal{A}^-(s) = \{a \in \mathcal{A}|\forall \tau \in \mathcal{T}(T(s,a)), \tau$ does not ends in $\mathcal{C}\}$. The intermediate rewards are given by a reasonable Process Reward Model (PRM) $r$. Terminal rewards $R$ indicate a correct solution ($R > 0$), and $0$ otherwise. A PRM is considered reasonable if the following conditions hold for all non-terminal states $s$: (i) $\forall a'_+ \in \mathcal{A}^+(s), \forall a'_- \in \mathcal{A}^-(s)$: $r(s,a^-) < r(s,a^+)$, and (ii) $\forall a_+ \in \mathcal{A}^+(s), \forall a_- \in \mathcal{A}^-(s), \forall \tau^- \in \mathcal{T}(T(s,a^-)), \forall \tau^+ \in \mathcal{T}(T(s,a^+))$: $\sum_{i \in \tau^-} r_i < \sum_{i \in \tau^+} r_i$. If these reasonable PRM conditions hold, then the greedy policy with respect to $Q^{\pi_u}$ defined by $\pi_{\mathrm{greedy}}(s) = \arg\max_a Q^{\pi_u}(s,a)$ always selects actions leading to a correct terminal state.*

*Proof.* For any state $s$, the state-action Q-function under the uniform random policy $\pi_u$ can be expressed as

$$Q^{\pi_u}(s, a) = r(s, a) + V^{\pi_u}(T(s, a)), \tag{13}$$

where $V^{\pi_u}(s) = \sum_{\tau \in \mathcal{T}(s)} p(\tau) \sum_{i \in \tau} r_i$.

For any $a^+ \in \mathcal{A}^+(s)$ and $a^- \in \mathcal{A}^-(s)$, the second PRM condition implies that $\forall \tau^- \in \mathcal{T}(T(s, a^-))$, $\forall \tau^+ \in \mathcal{T}(T(s, a^+))$: $\sum_{i \in \tau^-} r_i < \sum_{i \in \tau^+} r_i$. *[This condition ensures that the cumulative reward of any sub-trajectory starting from $T(s, a^-)$ (a child state leading to incorrect solutions only) is less than any sub-trajectory starting from $T(s, a^+)$ (a child state leading to at least a correct solution).]*

Therefore, we have that $V^{\pi_u}(T(s, a^-)) < V^{\pi_u}(T(s, a^+))$.

Based on the first PRM condition, we have that $r(s, a^-) < r(s, a^+)$. *(This condition ensures that actions maintaining at least a path to correct solutions receive higher intermediate rewards than those leading only to incorrect solutions.)*

Therefore, we obtain that

$$Q^{\pi_u}(s, a^-) = r(s, a^-) + V^{\pi_u}(T(s, a^-)) < r(s, a^+) + V^{\pi_u}(T(s, a^+)) = Q^{\pi_u}(s, a^+). \tag{14}$$

Thus, at any state $s$, $\pi_{\text{greedy}}(s) = \arg\max_a Q^{\pi_u}(s, a)$ will always select actions from $A^+(s)$ over actions from $A^-(s)$ (which means that it consistently choose actions that lead to states from which at least one sub-trajectory to a correct terminal state exists).

$\square$

Due to a reasonable PRM, by induction, following the greedy policy $\pi_{\text{greedy}}$ with respect to $Q^{\pi_u}$ always navigates to a correct solution, which guarantees its solution optimality.

## B  GRADIENT ANALYSIS

In this section, we analyze the relationship between our method and existing policy optimization methods from the gradient perspective.

**Proposition 1** *Assume only $\log \pi_\theta$ has parameters (i.e., LLM policy $\pi$ depends on $\theta$). Define importance sampling ratio* IS $= \frac{\pi_\theta(a|s)}{\pi_{\theta_{\text{old}}}(a|s)}$, *where $\pi_{\theta_{\text{old}}}$ is the behavior policy. Denote $\tilde{r}$ as our mean-centered reward. Then the gradient of ROVER's objective takes the following form, which is similar to policy-gradient:*

$$\nabla_\theta \mathcal{L}_{\text{ROVER}} = \mathbb{E}_{s,a,s' \sim P}\left[\left((\tilde{r} + Q') - \log \text{IS}\right) \nabla_\theta \log \pi_\theta(a|s)\right], \text{ where } Q' = \frac{1}{|\mathcal{V}|} \sum_{a' \in \mathcal{V}} Q(a'|s') \tag{15}$$

*Proof.* Recall the details of our method provided in § 3.2, ROVER has the following loss function:

$$\mathcal{L}_{\text{ROVER}} = \mathbb{E}_{s,a,s' \sim P}\left[\left(\tilde{r} + \frac{1}{|\mathcal{V}|} \sum_{a' \in \mathcal{V}} Q(a'|s') - Q(s, a)\right)^2\right]. \tag{16}$$

Let

$$u = \tilde{r} + \frac{1}{|\mathcal{V}|} \sum_{a' \in \mathcal{V}} Q(a'|s') - Q(s,a)$$

$$= \tilde{r} + \frac{1}{|\mathcal{V}|} \sum_{a' \in \mathcal{V}} Q(a'|s') - (\log \pi_\theta - \log \pi_{\theta_{\text{old}}})$$

$$= \tilde{r} + \frac{1}{|\mathcal{V}|} \sum_{a' \in \mathcal{V}} Q(a'|s') - \log\left(\frac{\pi_\theta}{\pi_{\theta_{\text{old}}}}\right) \tag{17}$$

$$= \tilde{r} + \frac{1}{|\mathcal{V}|} \sum_{a' \in \mathcal{V}} Q(a'|s') - \log \text{IS}.$$

Then the gradient is

$$\nabla_\theta \mathcal{L}_{\text{ROVER}} = \mathbb{E}_{s,a,s' \sim P}[u \cdot \nabla_\theta u] \tag{18}$$

Given that $Q' = \frac{1}{|\mathcal{V}|} \sum_{a' \in \mathcal{V}} Q(a'|s')$, where the gradient of $Q'$ is stopped (see Alg. 1), and $\pi_{\theta_{\text{old}}}$ does not involve gradient backpropagation, by combining Eq. 17 and Eq. 18 we have:

$$\nabla_\theta \mathcal{L}_{\text{ROVER}} = \mathbb{E}_{s,a,s' \sim P}\left[\left(\tilde{r} + \frac{1}{|\mathcal{V}|} \sum_{a' \in \mathcal{V}} Q(a'|s') - \log \text{IS}\right) \nabla_\theta \log \text{IS}\right]$$

$$= \mathbb{E}_{s,a,s' \sim P}\left[\left(\tilde{r} + \frac{1}{|\mathcal{V}|} \sum_{a' \in \mathcal{V}} Q(a'|s') - \log \text{IS}\right) \nabla_\theta \log \pi_\theta(a|s)\right]. \tag{19}$$

$$\square$$

Note the gradient of a typical policy optimization method, i.e., GRPO (Shao et al., 2024), is

$$\nabla_\theta \mathcal{L}_{\text{GRPO}} = \mathbb{E}_{s,a}[A \cdot \text{IS} \cdot \nabla_\theta \text{IS}] = \mathbb{E}_{s,a}[A \cdot \text{IS} \cdot \nabla_\theta \log \pi_\theta(a|s)]. \tag{20}$$

Therefore, we have the following key observation:

- Both gradients share the term $\nabla_\theta \log \pi_\theta$ (core of policy gradient).

- When importance sampling ratio $\text{IS} \to 1$ (small policy update), i.e., $\log \text{IS} \to 0$, so:

$$\nabla_\theta \mathcal{L}_{\text{ROVER}} \approx \mathbb{E}\big[(\tilde{r} + Q') \nabla_\theta \log \pi_\theta\big], \quad \nabla_\theta \mathcal{L}_{\text{GRPO}} = \mathbb{E}\big[A \cdot \nabla_\theta \log \pi_\theta\big].$$

These two objectives can be approximately equal if we remove the term $Q'$ in ROVER and the advantage $A$ in GRPO is normalized without the standard deviation term.

### B.1 EMPIRICAL JUSTIFICATION

**Ablation on the term $Q'$.** The Bellman target used for Q-value updates is composed of two components: centered reward, $\tilde{r}$, and the expected Q-value of the successor state under a uniform policy, $Q' = \frac{1}{|\mathcal{V}|} \sum_{a_{t+1} \in \mathcal{V}} Q(a_{t+1}|s_{t+1})$. We ablate the contribution of $Q'$ in the Bellman target by scaling it with a coefficient $\beta = [0.0, 0.2, 1.0, 5.0]$. The results show that this term is essential: removing it ($\beta = 0$) causes a collapse in entropy and response length (see Fig. 13(c) and 13(d)), leading to a sharp drop in pass@$k$ performance. Conversely, an overly dominant Q-term ($\beta = 5.0$) diminishes the reward signal, which also degrades performance. Crucially, as shown in Fig. 13(a) and 13(b), our method is not sensitive to the precise scaling of this term, with performance remaining stable across a wide range ($\beta$ from 0.2 to 1.0). By default, we set $\beta = 1.0$ in other experiments. Detailed pass@$k$ performances under different $\beta$ values are shown in Fig. 14.

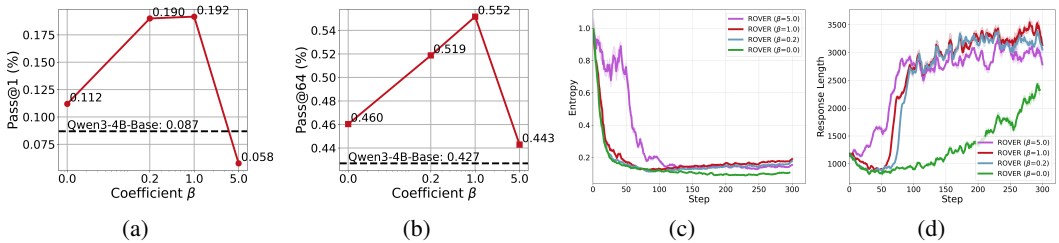

Figure 13: (a)&(b): Impact of coefficient $\beta$ in ROVER on pass@1 & pass@64, average performance on AIME24, AIME25, HMMT25 is reported. The X-axis is on a log scale. (c)&(d): Entropy and response length curves throughout training. All experiments are conducted on Qwen3-4B-Base with LLM decoding temperature $1.0$, and trained for 300 steps.

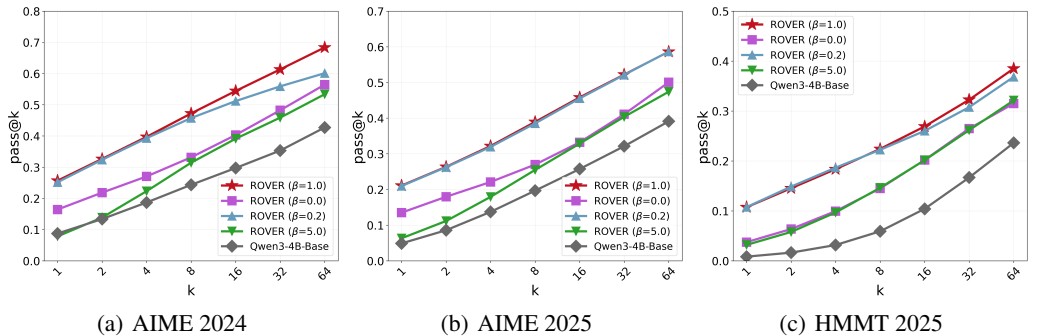

Figure 14: pass@$k$ performances under different value of coefficient $\beta$ in ROVER . All experiments are conducted on Qwen3-4B-Base and trained for 300 steps.

## C   GENERALIZABILITY OF ROVER

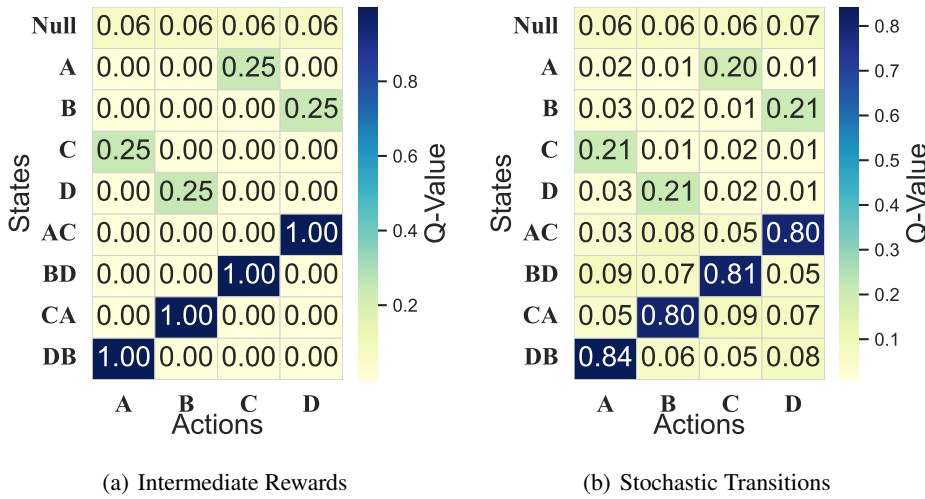

Figure 15: Q value maps of ROVER across different environmental settings within a tree-structured MDP. (a) ROVER generalizes well to environments with intermediate rewards, successfully learning optimal Q values. (b) ROVER also handles stochastic environments well, where the Q values of optimal actions are much larger than sub-optimal actions.

### C.1 GENERALIZING TO ENVIRONMENTS WITH INTERMEDIATE REWARDS

Since widely-used RLVR benchmarks basically follow the deterministic dynamic transitions with a terminal reward that indicates the correctness, we modify the didactic MDP in Fig. 5(a) in our paper to validate the practical effectiveness of ROVER for intermediate rewards and stochastic transitions. For intermediate rewards, we assign a reward of 1 to the intermediate states $\{A, B, C, D, AC, BD, CA, DB\}$ that can lead to the correct terminal state. In this environment, ROVER still captures all optimal solutions with a 100% success rate. We find that the Q value map learned by ROVER is optimal, as shown in Fig. 15.

### C.2 GENERALIZING TO ENVIRONMENTS WITH STOCHASTIC TRANSITIONS

Following the stochastic dynamics defined in environments like Atari (Machado et al., 2018), we modify the environment dynamics to be stochastic by setting $P(s+a|s,a) = 0.75, P(s+a'|s,a) = 0.25$, where $a'$ denotes the action different from the chosen action $a$. This means that there is a 0.25 probability for the agent to travel a random path, thereby ensuring the dynamic stochastic. Our empirical results demonstrate that ROVER can effectively handle this kind of stochasticity. If the dynamic becomes deterministic at test time, ROVER successfully captures all modes with a 100% success rate. The learned Q value map is still near-optimal, as shown in Fig. 15. If the dynamic remains stochastic at test time, ROVER achieves a 77% success rate, which is still high considering the 25% randomness in the environment.

### C.3 GENERALIZING TO ENVIRONMENTS WITH GRAPH-STRUCTURED MDP

To evaluate ROVER's generalizability to graph-structured environments, we insert a graph connection between $\{A, B, C, D\}$, which means a state in $\{A, B, C, D\}$ can have multiple parent nodes, leading to a graph-structured MDP. We provide an illustration of this graph structure in Fig 16. Our empirical results show that ROVER is still optimal under graph-structured MDPs, achieving a 100% success rate with all four modes covered.

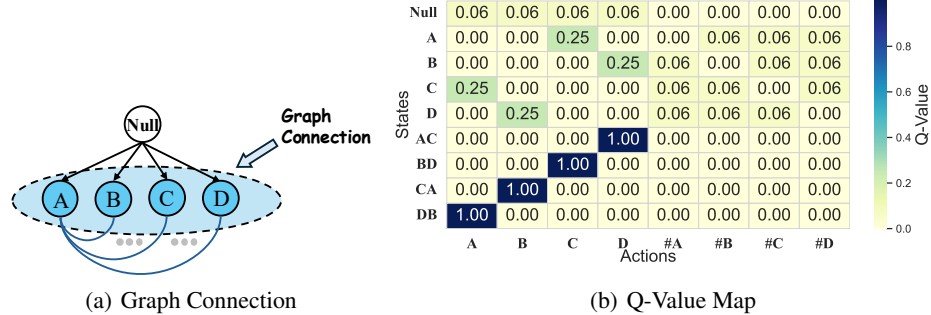

(a) Graph Connection                    (b) Q-Value Map

Figure 16: Q-value maps of ROVER on a graph-structured MDP. (a) States $\{A, B, C, D\}$ connect with each other, thereby making a graph-structured MDP. (b) '#A', '#B', '#C', '#D' denote the graph connection actions. We observe that ROVER still learns optimal Q-values under this graph structure.

## D RELATED WORK

RL with verifiable rewards (RLVR) (Guo et al., 2025; Team, 2025; Yang et al., 2025a; Comanici et al., 2025) has found great success in post-training LLMs on verifiable tasks. To bypass the need for the value model of PPO (Schulman et al., 2017), many actor-only variants have been proposed, such as GRPO (Shao et al., 2024), RLOO (Ahmadian et al., 2024), ReMax (Li et al., 2024), and RE-INFORCE++ (Hu et al., 2025a). Nevertheless, leading algorithms like GRPO still exhibit unstable learning dynamics and are prone to model collapse. Recent works propose to add various heuristics on advantage normalization (Liu et al., 2025c; Zheng et al., 2025; Zhao et al., 2025), clipping ratio (Yu et al., 2025), KL regularization (Liu et al., 2025a), entropy loss (He et al., 2025c; Zhang et al.,

2025a), reward bonous (He et al., 2025a; Li et al., 2025a; Yao et al., 2025; Cheng et al., 2025; Chen et al., 2025a;c), data augmentation (Yang et al., 2025b; Liang et al., 2025), and others (Cui et al., 2025; Wang et al., 2025). Crucially, these existing works are still constrained by the same surrogate, policy-gradient-based PPO objective, and often necessitate complex, case-specific tuning (Liu et al., 2025d). Our work departs from this paradigm, proposing a method grounded in random policy valuation that offers a minimalist yet theoretically guaranteed approach to fine-tuning LLMs.

Table 4: Summarization of MDP structures between different tasks, considering the discrete Atari task from traditional RL and the countdown task from RLVR. While traditional RL tasks have smaller spaces and shorter horizons, the underlying MDP structure can be much more complex than LLM RLVR tasks that feature deterministic, episodic, tree-structured MDPs (which have larger spaces and longer horizons and leverage a powerful pre-trained model that can navigate in the large space).

| Feature | Traditional RL Tasks (e.g., Atari) | LLM Reasoning Tasks (e.g., Countdown) |
|---|---|---|
| Transition dynamics | Stochastic/Deterministic | Deterministic |
| Reward function | Stochastic/Deterministic | Deterministic |
| | Intermediate/Episodic | Episodic |
| State space structure | Graph-like (often be cyclic) | Tree-like (no cycles) |
| Observability | Can be partial observable | Fully observable |
| Action space | Smaller | Larger |
| Horizon | Shorter | Longer |

## E  THE COUNTDOWN TASK

**Task Details.**  Countdown (Gandhi et al., 2025) is a math reasoning task capable of evaluating the arithmetic capabilities of LLMs. Below illustrates the toy example of the countdown task:

$$\text{nums:} \quad [19, 36, 55, 7],$$
$$\text{target:} \quad 65,$$
$$\text{answer:} \quad 55 + 36 - 7 - 19,$$

where the LLM should find the correct solution using the given numbers and basic arithmetic operations ($+, -, \times, \div$). The simplicity of the Countdown's reasoning path, yet challenging for small LLMs to solve effectively, makes it an accessible test bed for math reasoning.

**Training Details.**  We use the training and testing dataset provided by TinyZero (Pan et al., 2025). The training dataset contains 327680 problems, and the testing dataset contains 1024 unseen problems. A reward of 1 is given if the LLM finds the correct equation; Otherwise, it receives a zero reward. We set the batch size to 128 and the mini batch size to 64 during training. Optimization is conducted by an AdamW (Loshchilov & Hutter, 2017) where learning rate is $1 \times 10^{-6}$. The response length is set to 1k for both training and evaluation. We rollout 5 responses per prompt to calculate the mean-centered reward. Other configurations follow the default setting of TinyZero (Pan et al., 2025). For the baseline of GRPO with Clip_higher technique, we set clip ratio $\epsilon_{\text{low}} = 0.2$ and $\epsilon_{\text{high}} = 0.4$. Note that all the experimental settings across different methods remain the same for a fair comparison.

## F  RESULTS OF MATH TASKS ON DEEPSEEK-1.5B

We provide the details of the training setup on DeepSeek-R1-Distill-Qwen-1.5B model (Guo et al., 2025) as follows.

- We employ the datasets provided by DeepScaler (Luo et al., 2025), which contains 40k verifiable math questions.

- Built upon the veRL infra (Sheng et al., 2024), we set batch size to 128 and mini-batch size to 64.

Table 5: Results of DeepSeek-R1-Distill-Qwen-1.5B on typical math competition tasks. The high and the second-best scores are shown in bold and underlined, respectively.

| Models | Pass@1 | | | | Pass@64 | | | |
|---|---|---|---|---|---|---|---|---|
| | AIME24 | AIME25 | AMC23 | MATH | AIME24 | AIME25 | AMC23 | MATH |
| DeepSeek-R1-Distill-Qwen-1.5B (Guo et al., 2025) | 29.3 | 24.3 | 62.5 | 82.9 | 79.8 | 58.3 | 92.9 | 97.3 |
| DeepScaleR-1.5B (Luo et al., 2025) | 41.6 | 30.8 | 73.4 | 87.7 | 78.5 | 62.9 | 95.0 | 96.8 |
| ProRLv2-Qwen-1.5B (Liu et al., 2025a) | 52.6 | 35.2 | 81.5 | 90.6 | 79.2 | 59.7 | 94.3 | 96.1 |
| **ROVER (Ours)** | 42.2 | 31.2 | 74.3 | 88.3 | **80.6** | **64.4** | **95.2** | **97.1** |

- we use the AdamW (Loshchilov & Hutter, 2017) optimizer with a constant learning rate of $1 \times 10^{-6}$ for gradient backpropagation.

- We rollout 8 responses per prompt to calculate the mean-centered reward $\tilde{r}$.

- Following DeepScaler (Luo et al., 2025), we first train DeepSeek-R1-Distill-Qwen-1.5B for 1k steps with a 8k response length. Then we scale the response to 16k for an additional 1k training steps. Experiments are conducted on 8 H200 GPUs for around 5 days.

Following the evaluation scripts provided by veRL, we use a sampling temperature of 0.6, nucleus sampling (Holtzman et al., 2019) with top_p = 0.95, and a maximum response length of 24k for evaluation. We evaluate our ROVER-trained model and previous SOTA models such as DeepScaler-1.5B (Luo et al., 2025) and ProRLv2-1.5B (Liu et al., 2025a) on AIME24, AIME25, AMC23, and MATH tasks. We rollout 128 responses per prompt for each task, and report both the pass@1 (avg@128) and pass@64 (calculated by an unbiased estimator (Chen et al., 2021)) for comprehensive comparison.

From the results summarized in Table 5, we observe that ROVER achieves the best performance in terms of both pass@1 and pass@64 scores compared with DeepScaler, which is trained on the same dataset as ours. Note that the comparison with ProRLv2 is not fair since ROVER uses more than $3\times$ smaller datasets (40k (ours) vs. 136k (ProRLv2)). Moreover, the training of ROVER only lasts for around 960 GPU hours, while ProRLv2 is trained for 16k GPU hours. However, thanks to the better reasoning diversity brought by our method, ROVER can achieve higher scores than ProRLv2 on pass@64.

## G   RESULTS OF MATH TASKS ON QWEN MODELS

### G.1   TRAINING AND EVALUATION DETAILS

**Training Details.** To ensure a fair comparison, both ROVER and baselines are trained using the same learning rate, batch size, and training steps (see Table 6). We fix 600 training steps for ROVER and baselines. During each training step, $128 \times 8$ samples are involved to calculate gradients. The computational requirements are approximately 1,280 GPU hours for experiments initialized with Qwen3-8B-Base and 832 GPU hours for those with Qwen3-4B-Base.

**Evaluation Details.** Default hyperparameters of evaluation are summarized in Table 7. To compute average pass@1, we sample 256 independent runs for AIME24, AIME25, HMMT25, and AMC23 for comprehensive evaluation to reduce the variance introduced by the relatively small sizes of these benchmarks, while 10 runs are sufficient for the larger OlympiadBench, MATH500, and GPQA-diamond benchmark.

**Baselines.** We compare ROVER with the following baselines:

- GRPO (Shao et al., 2024): It employs a standard implementation framework with token-level mean aggregation loss, serving as a fundamental baseline for LLM reinforcement learning.

- DAPO (Yu et al., 2025): It extends GRPO by introducing several techniques to enhance LLM training efficiency. These include clip-higher, dynamic sampling, and overlong reward shaping. We set $\epsilon_{low} = 0.2$, $\epsilon_{high} = 0.28$.

Table 6: Default hyperparameters for RL training.

| Hyper-parameter | Value |
| --- | --- |
| Temperature | 0.6 |
| Response length | $8 \times 1024$ |
| Responses per prompt | 8 |
| Train batch size | 128 |
| Mini batch size | 32 |
| PPO_epoch | 1 |
| Learning rate | $1e-6$ |

Table 7: Default hyperparameters for evaluation.

| Hyper-parameter | Value |
| --- | --- |
| Temperature | 0.6 |
| Response length | $24 \times 1024$ |
| top_p | 0.95 |

**Prompt** Find the number of ways to place a digit in each cell of a 2x3 grid so that the sum of the two numbers formed by reading left to right is $999$, and the sum of the three numbers formed by reading top to bottom is $99$. The grid below is an example of such an arrangement because $8+991=999$ and $9+9+81=99$.\n\\[\\begin{array}{|c|c|c|} \\hline 0 & 0 & 8 \\\\ \\hline 9 & 9 & 1 \\\\ \\hline \\end{array}\\] Let's think step by step and output the final answer within \\boxed{}.

**Qwen3-8B-Base and GRPO's Strategy 1**

**...** Condition 1: The sum of the two numbers formed by reading left to right is 999. (100a + 10b + c) + (100d + 10e + f) = 999.

Condition 2: The sum of the three numbers formed by reading top to bottom is 99.

(10a + d) + (10b + e) + (10c + f) = 99. This leads to: a + d = 9; b + e = 9; c + f = 9 **...** From the top-bottom sum condition, we get: a + b + c = 8. According to **_the stars and bars theorem_**, the number of non-negative integer solutions to a + b + c = 8 where each variable is a digit is: \binom{8+3-1}{3-1} = \binom{10}{2} = \boxed{45}.

**Qwen3-8B-Base and GRPO's Strategy 2**

**...** The two equations are: 1. \[ (100a + 10b + c) + (100d + 10e + f) = 999 \] 2. \[ (10a + d) + (10b + e) + (10c + f) = 99 \]. Obtain equation A: 10a + 11d + e - c = 100 Substitute and simplify to get equation C: 10b + 11c + f = 10d - 1. For each \( d \) from 1 to 9: Find valid \( (b, c) \) pairs that satisfy \( 10d - 10 \leq 10b + 11c \leq 10d - 1 \).**...**
**_Pattern found: for $d$ from 1 to 9, each value of $d$ corresponds to $d$ valid combinations._**
Therefore, the total number is: **1 + 2 + 3 + ... + 9 = \boxed{45}.**

- - - - - - - - - - - - - - - - - - - - - - - - - - - - - - - - - - - - - - - - - - - - - - - - - - - - - - - - - - - - - - -

**ROVER's Strategy 1**

**...**
For a 2x3 grid, we have two key equations: the horizontal sum (100a + 10b + c) + (100d + 10e + f) = 999 and the vertical sum (10a + d) + (10b + e) + (10c + f) = 99.
**...**
Now we need to find the number of solutions to a + b + c = 8 where a, b, c are digits. This is **_a classic "stars and bars" problem_** with the constraint that a, b, c \leq 9. The number of solutions to \(a + b + c = 8\) in non-negative integers is given by: \binom{8 + 2}{2} = \binom{10}{2} = \boxed{45}.

**ROVER's Strategy 2**

The two equations are **...** This leads to three key equations: a + d = 9; b + e = 9; c + f = 9 For each value of c from 0 to 8: - f is determined by: \[ f = 9 - c \] - a + b must satisfy: \[ a + b = 8 - c \] - Number of valid (a,b) pairs equals (8-c+1)
**...**
**_Pattern found: for $c$ from 0 to 8, each value of $c$ corresponds to (9-$c$) valid combinations._**
Therefore, the total number is: **9 + 8 + 7 + ... + 1 = \boxed{45}.**

**ROVER's Strategy 3**

Let's denote **...**
- S_1 = a + d; S_2 = b + e; S_3 = c + f
- T_1 = a + b + c; T_2 = d + e + f

**...** Now, we need to find the number of non-negative integer solutions to \( d + e + f = 19 \) where \( 0 \leq d, e, f \leq 9 \). Using the **_inclusion-exclusion principle_**:

1. Total solutions without constraints: \( \binom{21}{2} = 210 \) 2. Solutions where one variable ≥ 10: \( 3 \times 55 = 165 \) Total valid solutions =
**210 - 165 = \boxed{45}.**

**ROVER's Strategy 4**

**...** Subtracting equation 2 from equation 1 and dividing by 9: - c = 10a + 11d + e - 100 Since \(c\) is a digit (0-9): - 100 \leq 10a + 11d + e \leq 109 Analyzing the possible values of a, d, e (all being digits 0-9), we discover that integer solutions satisfying the inequality **_exist only when a = 5, 6, 7, 8, 9._** For each value of a, there are **_exactly 9 corresponding combinations of (d,e)_**. Therefore: Total number of combinations =
**5 × 9 = \boxed{45}.**

Figure 17: Illustration of strategies discovered by Qwen3-8B-Base, GRPO and ROVER . Responses sharing the same color represent strategically identical approaches. While Qwen3-8B-Base and GRPO find two distinct strategies 1&2, ROVER not only discovers the same two strategies but also uncovers two additional strategies 3&4. For example, beyond discovering the *Stars and Bars theorem* (strategy 1), ROVER also discovered a solution based on the *inclusion-exclusion principle* (strategy 3), which demonstrates ROVER 's capability in pushing reasoning boundaries.

- REINFORCE++ (Hu et al., 2025a): Different from GRPO, it incorporates global advantage normalization (across responses correspond to different prompts within a batch), resulting in an unbiased approach that significantly improves training stability. We implement the REINFORCE++-baseline version in this paper.

All baselines are rigorously implemented following the official veRL recipes (Sheng et al., 2024).

### G.2 CASE STUDIES

**Discovered strategies comparison.** To intuitively show the enhanced diversity of ROVER , we present a representative prompt from AIME24 that holds multiple potentially feasible strategies. For each model, 32 samples are generated and subsequently clustered based on strategic equivalence using an LLM judge (the prompt of the LLM judge is given by Fig. 27). Representative CoT examples for each cluster are illustrated in Fig. 17.

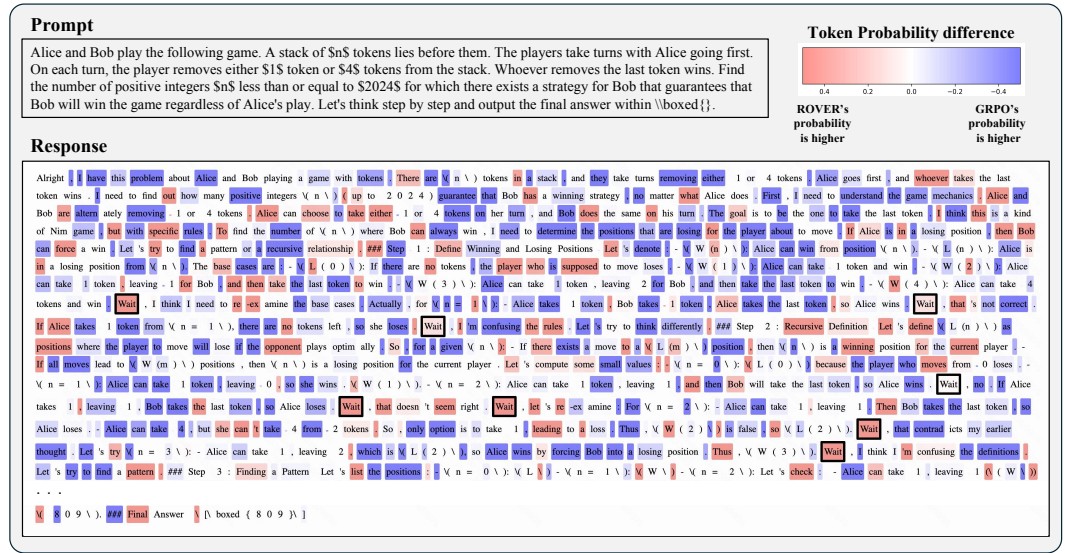

Figure 18: Token probability differences between ROVER and GRPO (visualized by the heatmap). ROVER exhibits a significantly higher probabilities to tokens associated with *reasoning contrasts or shifts*, exemplified by " Wait ".

**Token probability comparison**. We visualize a case study to show the token probability differences between ROVER and GRPO in Fig. 18 (a representative prompt from AIME24 is selected). ROVER demonstrates higher probabilities for tokens indicating *contrasts or shifts*, particularly "wait", which facilitates the exploration of alternative reasoning paths, thereby contributing to increased strategic diversity. The specific *forking tokens* mentioned in § 4 are shown in Table 8.

Table 8: *Forking token* categories and their corresponding tokens.

| Category | Tokens |
|---|---|
| mathematical_setup | suppose, assume, given, define |
| contrasts_shifts | wait, however, unless |
| progression_addition | thus, also |

### G.3 ADDITIONAL EXPERIMENT RESULTS

**Pass@$k$ results on Qwen3-4B-Base**. As shown in Fig. 19, similar to results on Qwen3-8B-Base, ROVER demonstrates consistently superior pass@$k$ performance on Qwen3-4B-Base across all $k$ values, while other RL baselines drop when $k$ becomes higher.

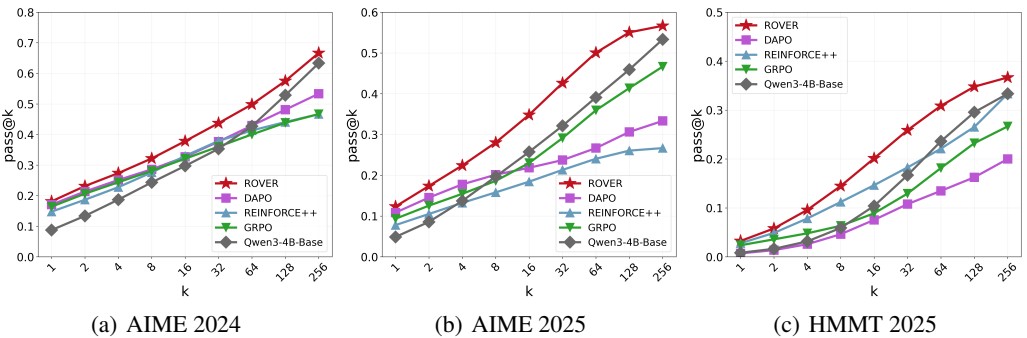

(a) AIME 2024          (b) AIME 2025          (c) HMMT 2025

Figure 19: pass@$k$ performances of ROVER and baselines (Qwen3-4B-Base).

**Maj@$k$ results.** The supplemental results of maj@$k$ performance on AIME25 for Qwen3-8B-Base is shown in Fig. 20. To mitigate random variations in evaluation results, we adopt a repeated sampling approach for computing maj@$k$: $k$ responses are randomly sampled from the response collection, and this sampling procedure is repeated 1000 times with the average value reported.

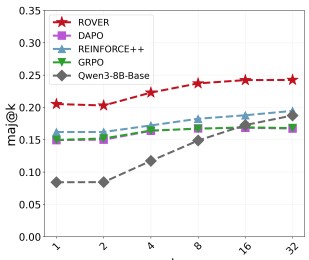

### G.3.1 ABLATION OF TEMPERATURE $\rho$

Consistent with the findings on the countdown task in Fig. 8, the training temperature $\rho$ serves as an exploration-exploitation trade-off. A large $\rho$ ($\rho = 4$) results in more stochastic behavior and constant entropy throughout training, which affects the performance (see Fig. 21). Conversely, a smaller $\rho$ ($\rho = 0.01$) leads to a greedy and deterministic policy, which compromises diversity (e.g., reduced pass@$k$) for improved pass@1 performance. By default, we set $\rho = 1$ in other experiments.

Figure 20: Maj@$k$ performance of ROVER and baselines on AIME25 for Qwen3-8B-Base.

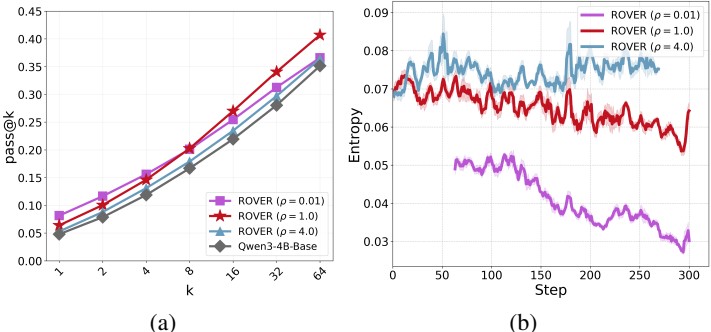

Figure 21: Impact of temperature $\rho$. All experiments are conducted on Qwen3-4B-Base and trained for 300 steps. (a): pass@$k$ results (average performance on AIME24, AIME25, HMMT25 are reported). (b): entropy curves throughout training.

### G.3.2 TRAINING DYNAMICS

We present the training curves of entropy in Fig. 22. The min, mean, and max values of $\tilde{r}$ and $Q'$ within a training batch are visualized in Fig. 23.

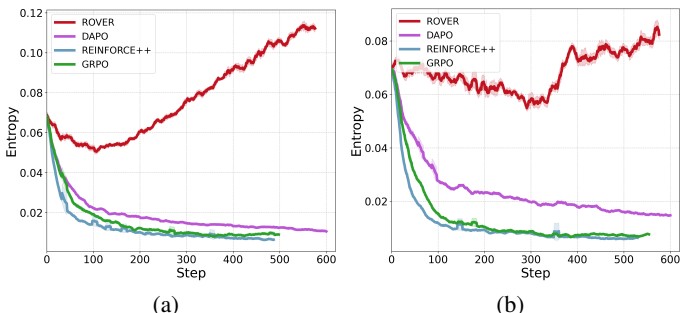

Figure 22: Training curves of entropy for ROVER and baselines. (a) & (b) are results on Qwen3-8B-Base and Qwen3-4B-Base, respectively. The entropy of ROVER is maintained at a relatively higher level, and can even increase stably at later training stages, indicating expanded exploration space. In contrast, the entropy of baselines inevitably decreases to a low level.

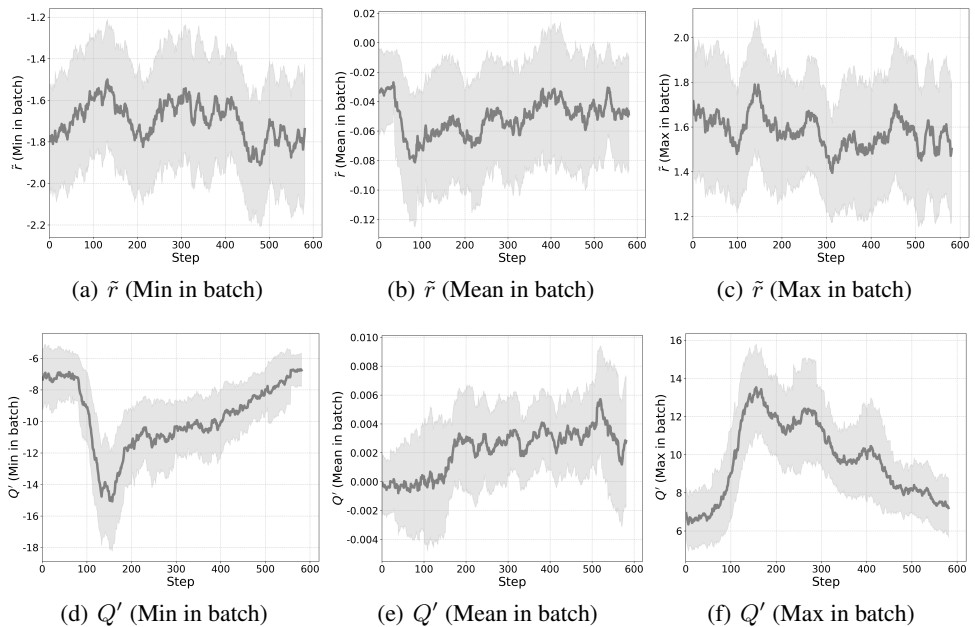

Figure 23: Absolute scales of $\tilde{r}$ and $Q'$ throughout training for ROVER (trained on Qwen-8B-Base).

### G.3.3 COMPARISON WITH ENTROPY REGULARIZATION METHODS

To investigate the effectiveness of ROVER's diversity-enhancing capability, we implement an entropy-aware variant of GRPO as a comparison. Specifically, the GRPO objective is augmented with an entropy term: $r'(s, a) = r(s, a) + \alpha\mathcal{H}(\pi(\cdot|s))$, where $\mathcal{H}$ denotes the policy entropy and $\alpha$ controls the strength of diversity encouragement. Following (Jiang et al., 2025), we set $\alpha = 0.005$ and train the Qwen3-4B-Base model to compare performance across methods.

As shown in Table 9, while the entropy bonus successfully maintains higher entropy levels, it leads to performance degradation in GRPO. In contrast, ROVER achieves a superior balance, simultaneously improving performance while preserving significantly higher entropy.

Table 9: Comparison with entropy regularization methods. Pass@1 is averaged over AIME24, AIME25, and HMMT25.

|  | Pass@1 | Entropy (step 300) |
| --- | --- | --- |
| Qwen3-4B-Base + GRPO | 9.40 | 0.008 |
| Qwen3-4B-Base + GRPO ($\alpha = 0.005$) | 9.31 | 0.020 |
| Qwen3-4B-Base + ROVER | **11.10** | **0.055** |

### G.4 MEASURING THE DIVERSITY OF LLM RESPONSES

In addition to the number of distinct strategies mentioned in § 4, we additionally incorporate two diversity metrics for a comprehensive evaluation. These diversity metrics are introduced as follows.

**No. of Distinct strategies** (Zhang et al., 2025c). It categorizes all generated responses into equivalent strategy classes and counts the total number of distinct classes.

**Utility** (Zhang et al., 2025c). It combines diversity and quality using a user patience model where users have a probability $p$ of requesting additional generations. It rewards novel responses while applying geometric decay to account for diminishing user attention over multiple generations. Models capable of generating multiple correct responses with distinct strategies will receive a higher utility score.

**Cosine Distance**. We embed all responses using Qwen3-8B-Embedding (Zhang et al., 2025b) and compute the average pairwise cosine distance between response vectors. Higher distances indicate greater semantic diversity among generated responses. Specifically, given a set of generated responses $\{y_1, y_2, \ldots, y_n\}$, let $E(y_i) \in \mathbb{R}^d$ denote the L2-normalized embedding vector of response $y_i$ obtained from Qwen3-8B-Embedding. The pairwise cosine similarity between responses $y_i$ and $y_j$ is:

$$S(y_i, y_j) = E(y_i) \cdot E(y_j).$$

The average pairwise cosine similarity is:

$$\bar{S} = \frac{1}{n(n-1)} \sum_{i=1}^{n} \sum_{j \neq i} S(y_i, y_j).$$

Finally, the cosine distance is defined as $1 - \bar{S}$.

As a supplement to Fig. 10, results of quality-diversity trade-off across $t \in [0.3, 0.9, 1.2]$ are shown in Fig. 24.

Furthermore, we demonstrate the comparison on all three diversity metrics under different decoding temperatures in Fig. 25. ROVER consistently exhibits greater diversity across all decoding temperatures.

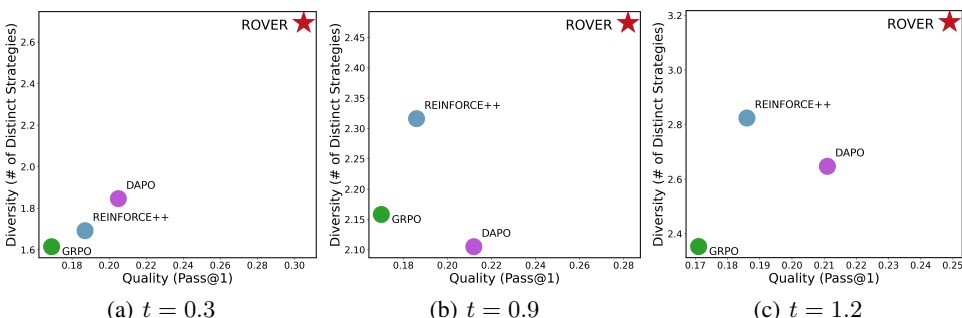

Figure 24: Quality-diversity trade-off with different decoding temperature (AIME24).

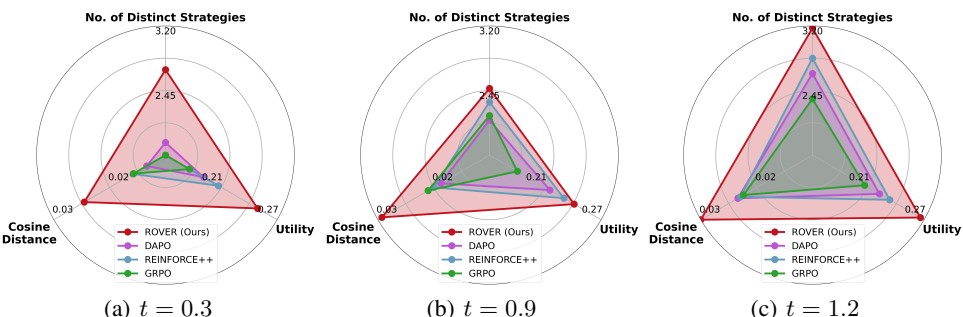

Figure 25: Comparison of multiple diversity metrics with different decoding temperatures (AIME24).

### G.5 PROMPTS

We present the prompt template for RL training and evaluation in Fig. 26, and the prompt for LLM judger in Fig. 27.

```
<|im_start|>user
{question}
Please reason step by step, and put your final answer within
\boxed{}.
<|im_end|>
<|im_start|>assistant
```

Figure 26: Prompt template for RL training and evaluation. The base model uses the same prompt template as the trained model during evaluation.

## H   RESULTS OF MATH TASKS ON LLAMA MODELS

To demonstrate the broad applicability of ROVER across different model architectures, we extend our experiments beyond the Qwen family by training Llama3.1-8B-Instruct on the hard training dataset from simpleRL-zoo Zeng et al. (2025), which consists of level 3-5 problems from the MATH training set. As shown in Table 10, ROVER achieves an avg@5 accuracy of 67.8% on MATH500, outperforming GRPO by 8.8%. These results further validate the generalizability of ROVER to diverse base models.

Table 10: Performance comparison on MATH500 when using Llama3.1-8B-Instruct.

|  | MATH500 (avg@5) |
| --- | --- |
| Llama3.1-8B-Instruct + ROVER | **67.8** |
| Llama3.1-8B-Instruct + GRPO | 59.0 |
| Llama3.1-8B-Instruct | 51.9 |

## I   RESULTS ON TOOL-USE TASKS

**Discussions about the MDP of the Tool-Use Domain.**   Regarding tool use, it is crucial to distinguish between the stochasticity inherent in external tools (e.g., a search engine $\mathcal{R}$) and the determinism of the state update mechanism $\mathcal{P}$. As illustrated in Fig. 28, the output generated by a tool acts as an *observation* from the environment. Once this observation is realized and received by the agent, it becomes a fixed component of the context history. Consequently, the transition function $\mathcal{P}(s_{t+1}|s_t, a_t)$, defined as the concatenation of the current state and the new token sequence, remains strictly deterministic ($\mathcal{P} = 1$). This structural property ensures that our theoretical framework holds even in scenarios involving stochastic tool execution.

In addition to reasoning tasks, we further validate ROVER in tool-use scenarios based on the Search-R1 Jin et al. (2025) setup. Specifically, we fine-tune the Qwen2.5-3B-Base model on the combined training sets of Natural Questions (NQ) and HotpotQA. Due to resource and time constraints, we evaluated the ROVER checkpoint at step 300. Although this falls short of the 500 steps used for GRPO in the original Search-R1, it allows for a meaningful comparison with the reported GRPO results.

**Performance comparison.** Table 11 presents the performance across multiple benchmarks, spanning both General QA and Multi-hop QA tasks. The results show that ROVER not only works effectively with tools but also outperforms the original GRPO in terms of average performance by 1.8%.

**Diversity comparison.** To evaluate diversity, we generate 32 responses for each prompt from the Bamboogle benchmark with temperature set to 1.0, using the cosine distance between agent responses as the diversity metric. As shown in Table 12, ROVER demonstrates substantially better diversity.

We hypothesize that diversity is particularly crucial for tool-use tasks and represents a promising future direction for our ROVER algorithm. An agent capable of generating diverse queries can

```
You are given the original prompt and two model-generated
responses. Determine whether the two responses use different
strategies to solve the problem.

Use the following guidelines to identify different strategies:

1. Mathematical Tools and Concepts:
- Using different mathematical tools (e.g., differentiation vs.
integration, series expansion vs. direct computation)
- Applying different theorems or properties (e.g., mean value
theorem vs. fundamental theorem of calculus)
- Different mathematical domains (e.g., algebraic vs. geometric,
analytical vs. combinatorial)

2. Solution Structure Differences:
- Different variable substitutions or transformations
- Different equation setups for the same problem
- Different ways of breaking down the problem into subproblems

3. Specific Examples of Different Approaches:
- Direct computation vs. recursive method
- Forward solving vs. backward solving (working from the answer)
- Algebraic manipulation vs. numerical approximation
- Using contradiction vs. direct proof
- Using induction vs. direct formula
- Coordinate-based vs. coordinate-free methods

Even if two solutions arrive at the same answer, they should be
considered different if they:
- Use different key mathematical tools or theorems
- Follow different logical sequences in critical steps
- Represent the problem using different mathematical frameworks
- Break down the problem in substantially different ways

Original prompt: {prompt}
Generation 0: {generation0}
Generation 1: {generation1}

Question: Do Generation 0 and Generation 1 use different
strategies? First analyze the key mathematical tools and
solution structure used in each solution, then respond
with "[[yes]]" if the generations use different
strategies or "[[no]]" if they do not.
```

Figure 27: Prompt for LLM judger to determine whether two responses use different strategies. We refined the prompt proposed in (Li et al., 2025a) to enhance the LLM judge's capability for more nuanced strategy classification.

retrieve more varied and comprehensive information from the corpus, thereby increasing its chances of discovering the key evidence required to answer a prompt correctly. Conversely, a model lacking diversity may fall into monotonous query patterns, leading to a failure loop where it repeatedly fetches the same unhelpful results from the search engine and fails to solve the problem.

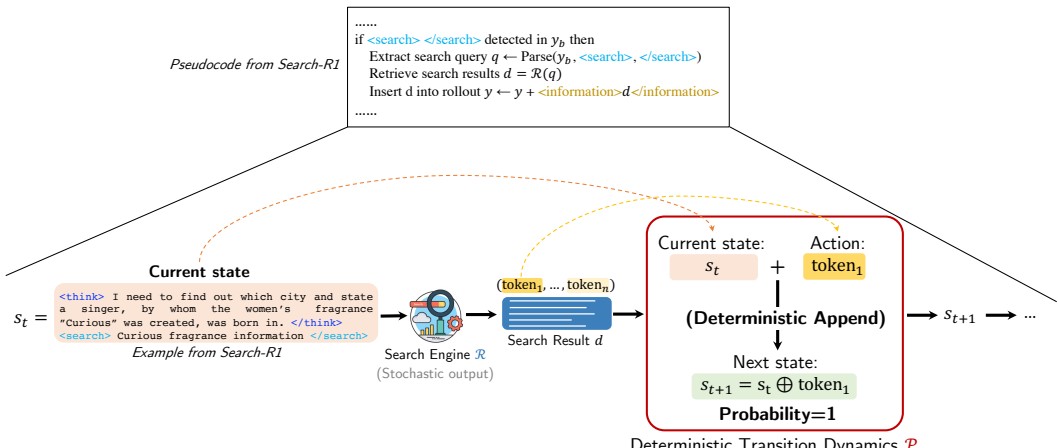

Figure 28: Illustration of the transition dynamics in the tool-use domain (e.g., Search-R1 (Jin et al., 2025)). The search result is modeled as a stochastic observation from the environment rather than a probabilistic component of the agent's transition function. Thus, while the content of search results may vary due to the stochastic nature of the search engine, the state transition logic (**concatenation**) remains deterministic.

Table 11: Performance comparison on General QA and Multi-hop QA benchmarks.

| | General QA | | | Multi-hop QA | | | Avg. |
|---|---|---|---|---|---|---|---|
| | NQ | TriviaQA | PopQA | HotpotQA | 2wiki | Musique | |
| Search-R1-base (GRPO) | 0.421 | 0.583 | 0.413 | **0.297** | 0.274 | 0.066 | 0.342 |
| ROVER | **0.477** | **0.600** | **0.443** | 0.290 | **0.286** | **0.069** | **0.360** |

## J    LLM USAGE DETAILS

In compliance with ICLR 2026 policies on large language model usage, we disclose that LLMs are mainly used for two purposes in this work:

- LLM Judge for Strategic Equivalence Assessment: We employed LLMs as judges to determine whether two model responses are strategically identical (see Fig. 27). This constitutes a core component of our research methodology. We have carefully validated the safety and reliability of the LLM judge outputs through systematic verification procedures.

- Writing Enhancement: We utilized LLMs to polish the paper's writing at the syntactic and grammatical levels. All LLM-generated content has undergone thorough human review and verification to ensure accuracy, appropriateness, and compliance with academic standards.

All LLM outputs were subject to careful human oversight and validation. We take full responsibility for the accuracy and integrity of all content in this paper, including any sections enhanced with LLM assistance.

Table 12: Diversity comparison evaluated on Bamboogle benchmark using cosine distance between agent responses.

| Method | Cosine Distance |
|---|---|
| Search-R1-base (GRPO) | 0.042 |
| ROVER | **0.065** |

