# OpenReview forum: "Random Policy Valuation is Enough for LLM Reasoning with Verifiable Rewards"
_ICLR.cc/2026/Conference — ICLR 2026 Poster_

### Official Review · Reviewer_PNvP · 2025-10-30

**Soundness:** 3
**Presentation:** 3
**Contribution:** 3
**Rating:** 6
**Confidence:** 4

**Summary:**

This paper presents ROVER, a simplified reinforcement learning algorithm for improving the reasoning abilities of large language models. The authors argue that standard methods like PPO are overly complex for the specific structure of math reasoning tasks, which they model as deterministic, tree-structured Markov Decision Processes. The core insight is that an optimal policy can be derived directly by evaluating a fixed, uniform random policy, thus avoiding the unstable iterative loops of generalized policy iteration. ROVER implements this by learning Q values based on a random policy and using a softmax sampler to promote diversity. Experiments show that ROVER significantly outperforms strong baselines in both solution quality and diversity.

**Strengths:**

1. The paper proposes an elegant simplification of the reinforcement learning process for this specific problem domain. By identifying the unique properties of the math reasoning MDP, the authors justify moving away from complex generalized policy iteration.

2. The empirical results are consistently strong and well presented. The proposed method, ROVER, not only achieves higher scores on pass@1 metrics but also shows substantial gains in diversity and pass@k at larger k values.

3. The work offers a valuable conceptual contribution by reframing the problem. The key insight is the mismatch between general-purpose RL algorithms and the specialized structure of LLM reasoning tasks. This perspective is insightful.

**Weaknesses:**

1. There is a gap between the theoretical foundation and the practical implementation of the Q function. The theory relies on the true Q values of a uniform random policy. However, the algorithm approximates this with a relative Q function based on the log probabilities of the current and previous policies. The paper does not provide a clear justification for why this specific practical choice is a good proxy for the theoretical quantity.

2. While the baselines are relevant, the comparison could be strengthened by including methods that explicitly aim to improve diversity through mechanisms other than softmax sampling. For instance, algorithms that incorporate an entropy bonus in the reward function or use other diversity-seeking objectives.

3. The novelty of the algorithmic components in the practical implementation is somewhat limited. The use of a relative Q function, mean-centered rewards, and reward broadcasting are techniques inspired by or used in prior works like GRPO and REINFORCE++. The main novelty lies in the theoretical motivation for combining them in this simplified framework, not in the components themselves.

**Questions:**

1. Could the authors discuss how you expect ROVER's performance to be affected in environments that violate the assumptions, such as tasks involving tool use with a chance of failure or those with partial credit rewards?

2. Algorithm 1 presents the loss function for updating the model. Could the authors please provide the explicit mathematical formula for the stop gradient target used in the L_ROVER calculation? The description in lines 295 and 296 is slightly ambiguous about how the target for the Q value regression is constructed from the next state.

3. The paper shows superior diversity compared to PPO-style methods. Did the authors consider comparing ROVER to other approaches that explicitly encourage diversity, for example, by adding a learned or fixed entropy bonus to the model's reward?

---

> ### Author Response · Authors · 2025-11-21
> **Response to Reviewer PNvP (Part 1)**
>
> We thank the reviewer for the careful reading and valuable feedback of our work, and we greatly appreciate the comments!
>
> > W1: There is a gap between the theoretical foundation and the practical implementation of the Q function. The theory relies on the true Q values of a uniform random policy. However, the algorithm approximates this with a relative Q function based on the log probabilities of the current and previous policies. The paper does not provide a clear justification for why this specific practical choice is a good proxy for the theoretical quantity.
>
> Thanks for the question. Line 8 in Algorithm 1 is indeed estimating the Q-value of the uniform random policy, which follows the standard definition of policy evaluation, i.e., $Q(s,a) = R(s,a) + \frac{1}{|A(s')|} \sum_{a' \in A(s')} Q(s', a')$ (in the context of math reasoning with discount factor $\gamma=1$). A trivial way is to directly train an addition Q-model from scratch, but this will incur sustantial costs. Instead, we require an efficient parameterization of the Q-function in practice. **We leverage the relationship between the policy $\pi$ and Q-values $Q$ in Theorem 2 to obtain an efficient representation of Q-model from the policy model.** Given the soft sampling based on the random policy's Q-values in ROVER (i.e., $\pi(a|s) \propto \exp(Q(s,a)/\rho)$ with $\rho$ the temperature), we can express $Q^u$ by log probabilities of the current policy: $Q(s,a) = \rho \log \pi(a|s)$ (where we choose to omit a state-dependent constant term that does not affect action selection), allowing for an efficient representation of our Q-model. In our practical implementation of ROVER, we introduce a reference point from the old policy to further stabilize the Q-model representation using $Q(s,a)=\rho (\log \pi_(a|s) - \log \pi_{\text{old}}(a|s))$, which centers the initial Q-values around zero and focuses updates on relative improvements rather than absolute values. This provides a more stable optimization point when scaling ROVER to extremely large action spaces, while still maintaining the core idea of random policy evaluation. Our extensive experiments confirm the effectiveness of this parameterization choice.
>
> > W2: While the baselines are relevant, the comparison could be strengthened by including methods that explicitly aim to improve diversity through mechanisms other than softmax sampling. For instance, algorithms that incorporate an entropy bonus in the reward function or use other diversity-seeking objectives.
>
> Thanks for the suggestion. To investigate the effectiveness of entropy bonus, we implemented entropy-aware GRPO [1], where the objective is augmented with an entropy term: $r'(s,a) = r(s,a) + \alpha \mathcal{H}(\pi(\cdot|s))$, where $\mathcal{H}$ represents policy entropy and $\alpha$ is a coefficient controlling the strength of the diversity encouragement. Specifically, we consider $\alpha = 0.005$ from [1]. We trained the Qwen3-4B-Base model and compared performance across methods. While the entropy bonus is able to maintain entropy levels (see the table below), which is also a widely adopted technique in maintaining diversity in classic RL, it leads to some performance degradation in GRPO (we believe the primary reason is due to the vast action space when given powerful pre-trained models, which is different from classic RL problems with smaller action space and without a powerful pre-trained policy model -- requiring the agent to explore as many states as possible). In contrast, ROVER achieves a better balance between performance improvement and entropy preservation.
>
> |               | Pass@1 (average on AIME24, AIME25 and HMMT25) | Entropy at training step 300 |
> | :-- | :- | --|
> | Qwen3-4B-Base + GRPO                          | 9.40                                          | 0.008                        |
> | Qwen3-4B-Base + GRPO ($\alpha$=0.005) | 9.31                                          | 0.020                        |
> | Qwen3-4B-Base + ROVER                         | **11.1**                                      | **0.055**                    |

---

> ### Author Response · Authors · 2025-11-21
> **Response to Reviewer PNvP (Part 2)**
>
> > W3: The novelty of the algorithmic components in the practical implementation is somewhat limited. The use of a relative Q function, mean-centered rewards, and reward broadcasting are techniques inspired by or used in prior works like GRPO and REINFORCE++. The main novelty lies in the theoretical motivation for combining them in this simplified framework, not in the components themselves.
>
> Our primary contribution is indeed theoretical, where we provide a novel perspective and establish that in deterministic tree-structured MDPs (which closely model LLM reasoning with verifiable rewards), the Q-function of the uniform random policy contains crucial information for finding optimal solutions directly without complex policy iteration methods. **This insight represents a significant advance in classic RL literature, challenging the conventional view that uniform policies are merely uninformative starting points and providing novel insights.** Regarding practical algorithm design, `ROVER's contribution lies in the realization of a simple yet effective RL algorithm for LLM reasoning with verifiable rewards,` eliminating unnecessary complexity that often makes deployment challenging, as popular methods like PPO were originally designed for control tasks with different problem properties.
>
> > Q1: Could the authors discuss how you expect ROVER's performance to be affected in environments that violate the assumptions, such as tasks involving tool use with a chance of failure or those with partial credit rewards?
>
> Thanks for the insightful question! While our original theoretical framework is motivated by standard properties in most RLVR benchmarks, we have extended our analysis to accommodate the scenarios that the reviewer mentioned. The following theoretical extensions and empirical validations in tool use further demonstrate that ROVER is robust across a wide range of real-world applications.
>
> ### Additional theoretical results and empirical validation with intermediate rewards
>
> Although current benchmarks typically use outcome-based terminal rewards, we have extended our theoretical results to accommodate intermediate feedback (detailed proof in Appendix A.3):
>
> `Theorem 3`: Consider a finite-horizon MDP with deterministic transitions and a tree-structured state space. Let $\pi_u$ be a uniform policy, $Q^{\pi_u}$ denote its Q-function, and the discount factor $\gamma=1$. For any non-terminal state $s$, define $\mathcal{C}$ to be the set of terminal states with correct rewards, $T(s,a)$ the next state reached by taking action $a$ from state $s$, and $\mathcal{T}(s)$ the set of all possible trajectories from state $s$ to terminal states. Define two disjoint subsets of actions at state $s$, where $\mathcal{A}^+(s)=\{a \in \mathcal{A} | \exists \tau \in \mathcal{T}(T(s,a))\ {\rm such\ that}\ \tau\ {\rm ends\ in}\  \mathcal{C}\}$, and $\mathcal{A}^-(s)=\{a \in \mathcal{A} | \forall \tau \in \mathcal{T}(T(s,a)), \tau\ {\rm does\ not\ ends\ in}\ \mathcal{C}\}$. The intermediate rewards are given by a reasonable Process Reward Model (PRM) $r$. Terminal rewards $R$ indicate a correct solution ($R > 0$), and $0$ otherwise. A PRM is considered reasonable if the following conditions hold for all non-terminal states $s$: (i) $\forall a_{+}^{'} \in \mathcal{A}^+(s), \ \forall a_{-}^{'} \in \mathcal{A}^-(s)$: $r(s, a^-) < r(s, a^+)$, and (ii) $\forall a_{+} \in \mathcal{A}^+(s), \ \forall a_{-} \in \mathcal{A}^-(s), \ \forall \tau^- \in \mathcal{T}(T(s,a^-)), \ \forall \tau^+ \in \mathcal{T}(T(s,a^+))$: $\sum_{i \in \tau^-} r_i < \sum_{i \in \tau^+} r_i$. If these reasonable PRM conditions hold, then the greedy policy with respect to $Q^{\pi_u}$ defined by $\pi_{\rm greedy}(s) = \arg\max_a Q^{\pi_u}(s,a)$ always selects actions leading to a correct terminal state.
>
> This extension demonstrates that our theoretical framework and algorithm remain effective with partial credit rewards, as might be provided by process reward models. Importantly, these assumptions are quite mild: if a process reward model violates these assumptions, it would be considered "unreasonable" and would cause issues for any reward-maximization RL method.
>
> Our empirical validation on the MDP in Figure 5(a) in the main text confirms that ROVER captures all optimal solutions with a 100\% success rate when given partial credit rewards (`as detailed in Appendix C.1`).

---

> > ### Author Response · Authors · 2025-11-21
> > **Response to Reviewer PNvP (Part 3)**
> >
> > ### Additional experiments with tool use
> > **(Why Tool Use Should Work with ROVER)** Regarding tool use, a key insight is that even with tools that appear stochastic (e.g., search engines), the MDP with token-level action definition still has deterministic transition dynamics. Consider a search tool example in Search-R1 [2] (or similar tool-use setups): when an LLM queries a search engine within its reasoning process, the search results may appear "random" or unpredictable. However, based on the MDP formulation with token-level action, the search result (whether successful or failed) becomes part of the state once received, and the next token generation is deterministic conditioned on this state.
> >
> > **Example**: Given a query "... \</search>", if the search engine responds "This is 1", then:
> > P(next_state="... \</search>\<information> This" | state="... \</search>\<information>", action="This")=1. Similarly, for a failed search (with response "Fail"):
> > P(next_state="... \</search>\<information> Fail" | state="... \</search><\information>", action="Fail")=1
> >
> > The key insight is that in the MDP (with token-level action definition), the agent's actions are individual tokens it generates, not the API calls themselves. The source of uncertainty (external tool) becomes fixed information in the state (analogous to the random outcomes and deterministic mappings in random variables). This applies to various LLM reasoning scenarios with verifiable rewards (e.g., Search-R1 [2]).
> >
> > To better address the reviewer's concern, **we conduct additional experiments to validate ROVER in tool-use scenarios based on the open-sourced Search-R1 [2] setup**. Given limited time and resources during the rebuttal phase, we evaluate ROVER trained with only 300 steps (lower than 500 steps of GRPO in the original Search-R1 paper [2]). However, ROVER outperforms the baseline by an obvious margin, which demonstrates its great potential in tool-use experiments.
> >
> > **(Performance comparison)** The performance across multiple benchmarks, spanning both General QA and Multi-hop QA, is presented in the table below, which **shows that ROVER not only works effectively with tools but actually outperforms the original GRPO in terms of average performance (even without extensive hyperparameter tuning of ROVER)**:
> >
> > |   Method  | NQ        | TriviaQA  | PopQA     | HotpotQA | 2wiki     | Musique   | Avg.      |
> > | --- | --- | --------- | --------- | -------- | --------- | --------- | --------- |
> > | Search-R1-base (GRPO) | 0.421     | 0.583     | 0.413     | 0.297    | 0.274     | 0.066     | 0.342     |
> > | ROVER                 | **0.477** | **0.600** | **0.443** | 0.290    | **0.286** | **0.069** | **0.360** |
> >
> >
> > **(Diversity comparison)** To evaluate diversity, we generated 32 responses for each prompt from the Bamboogle benchmark (with the temperature set to 1.0). We then use the cosine distance between agent responses as the diversity metric. The results in the table below indicate that ROVER shows better diversity.
> >
> > |                       | cosine distance (evaluated on Bamboogle) |
> > | :-------------------- | :---- |
> > | Search-R1-base (GRPO) | 0.042                                                 |
> > | ROVER                 | **0.065**                                                 |
> >
> > We argue that diversity is crucial for tool-use tasks, which also represents a potential future direction for our simple yet effective ROVER algorithm. An agent capable of generating diverse queries can retrieve more varied and comprehensive information from the corpus, thereby increasing its chances of discovering the key evidence required to correctly answer a prompt. Conversely, a model lacking diversity may fall into monotonous query patterns, leading to a failure loop where it repeatedly fetches the same unhelpful results from the search engine and is unable to solve the problem.
> >
> > **In summary, these additional theoretical analyses and experimental results demonstrate that ROVER's simplicity and effectiveness, which also extend naturally to tool-augmented reasoning tasks, further validate our approach across diverse application domains.**

---

> > > ### Author Response · Authors · 2025-11-21
> > > **Response to Reviewer PNvP (Part 4)**
> > >
> > > > Q2: Algorithm 1 presents the loss function for updating the model. Could the authors please provide the explicit mathematical formula for the stop gradient target used in the L_ROVER calculation? The description in lines 295 and 296 is slightly ambiguous about how the target for the Q value regression is constructed from the next state.
> > >
> > > Thanks for your question! The explicit mathematical formula (lines 295-296) can be written as $\mathcal{L}\_{\text{ROVER}} = \frac{1}{\sum\_{i=1}^n |y\_i|} \sum\_{i=1}^n \sum\_{t=0}^{|y_i|-1} \left(Q(a_t|s_t) - \text{sg}[\hat{Q}(a_t|s_t]\right)^2$, where $\hat{Q}(a_t|s_t) = \tilde{r} + \frac{1}{|\mathcal{V}|} \sum_{a_{t+1}\in\mathcal{V}} Q(a_{t+1}|s_{t+1})$ represents the target Q-value for the uniform random policy computed via one-step backu. The stop gradient operation $\text{sg}[\hat{Q}(a_t|s_t)]$ means that the current Q-value estimate $Q(a_t|s_t)$ is updated to match the target $\hat{Q}(a_t|s_t)$, but the target itself does not receive gradient updates. During backpropagation, gradients flow only through $Q(a_t|s_t)$). This avoids the moving target problem [3] in the classic RL literature, which could otherwise lead to instability, ensuring that the target values remain stable during optimization. In addition, we also provided a comprehensive gradient analysis of ROVER's objective in Appendix B.
> > >
> > > > Q3: The paper shows superior diversity compared to PPO-style methods. Did the authors consider comparing ROVER to other approaches that explicitly encourage diversity, for example, by adding a learned or fixed entropy bonus to the model's reward?
> > >
> > > Please see our responses to W2, where we further validate ROVER's effectiveness by comparing it against the additional baseline with an entropy bonus in the reward.
> > >
> > > **Reference**
> > >
> > > [1] Jiang, Yuxian, et al. "Rethinking entropy regularization in large reasoning models." arXiv preprint arXiv:2509.25133 (2025).
> > >
> > > [2] Jin, Bowen, et al. "Search-r1: Training llms to reason and leverage search engines with reinforcement learning." arXiv preprint arXiv:2503.09516 (2025).
> > >
> > > [3] Van, Hasselt, et al., Deep reinforcement learning with double Q-learning, AAAI 2016
> > >
> > > ---
> > >
> > > **We thank the reviewer for the time and effort in reviewing our work! We would very much appreciate it if the reviewer could check our responses and the updates in the paper and let us know whether they address your concerns. We are happy to provide further clarification if you have any additional concerns.**

---

> ### Author Response · Authors · 2025-11-26
> **Looking forward to your reply!**
>
> Dear Reviewer PNvP,
>
> We hope this message finds you well. We are writing to kindly follow up on our response to your valuable comments and questions, which we submitted over 5 days ago.
>
> In our rebuttal, we addressed your comments by clarifying the design principle of our Q function, adding new experiments to compare to a new baseline with an entropy bonus [1] (`see Appendix G.3.3`), adding new experiments on the tool-use domain (`see Appendix I`), a new theorem and additional experiments to demonstrate the generalizability of our method to other environments (`see Appendix A.3 and Appendix C`), and detailed clarifications to illustrate the novelty of our method.
>
> We believe that we have clarified the concerns and questions raised. If you have any additional thoughts or questions about our responses, we would be more than happy to address them.
>
> Thank you once again for your insightful review! We look forward to your further feedback.
>
> Best,
>
> The authors
>
> [1] Jiang, Yuxian, et al. "Rethinking entropy regularization in large reasoning models." arXiv preprint arXiv:2509.25133 (2025).

---

> > ### Comment · Reviewer_PNvP · 2025-11-27
> > **Thanks for your detailed response.**
> >
> > Thanks for the detailed response. Overall, the response has strengthened my view of the paper.
> >
> > By the way, I have little caution against the theoretical argument that tool-use MDPs are deterministic because the outcome becomes part of the state. I hope the authors can further improve it and make it clearer.

---

> ### Author Response · Authors · 2025-11-27
> **Thanks for your reply!**
>
> Dear Reviewer PNvP,
>
> Thank you for this insightful comment and for prompting us to clarify this crucial theoretical point, as the distinction is subtle but fundamental. We have added an illustrative figure in `Appendix I` to visualize this process using Search-R1 [1] as an example, and we offer the following more detailed explanation.
>
> Specifically, it is crucial to distinguish between the randomness of the external tool (e.g., a search engine $\mathcal{R}$) and the determinism of the state update mechanism $\mathcal{P}$. Our argument hinges on the precise definition of the MDP components within the context of LLM reasoning with verifiable rewards.
>
> - **State ($s_t$)**: The state is the partial sequence of tokens generated so far, i.e., $s_t = ({\rm token}_1, ..., {\rm token}_t$).
> - **Action ($a_t$: token-level)**: The action is the next single token generated $a_t = {\rm token}_{t+1}$ (to be appended to $s_t$).
> - **Transition Function ($\mathcal{P}(s_{t+1}|s_t, a_t)$)**: The transition function maps the current state (partially generated seq) and an action (a single token) to the next state. Given the definitions, the next state $s_{t+1}$ is simply the `concatenation` of the current state $s_t$ and the next token $a_t$ (in tool use scenarios, the token is obtained from the search results). Therefore, $\mathcal{P}(s_{t+1} = {\rm concat}(s_t, a_t) | s_t, a_t) = 1$. **This transition is, by definition, deterministic, which is different from traditional RL tasks like computer games.** Please note that ROVER's theoretical framework relies solely on the deterministic nature of the state update mechanism, and can be applied in tool use scenarios with stochasticity in the external tool call processes.
>
> Thus, in Search-R1 (and standard RLVR with tool use practices), when a search query $q$ is detected, the search engine $\mathcal{R}$ is invoked (as shown in the pseudocode below from Search-R1), where the search result $d \sim \mathcal{R}(q)$ can be stochastic. While the search results may vary due to stochasticity in $\mathcal{R}$, the transition logic (concatenation) $\mathcal{P}$ remains deterministic (which is different from stochasticity in transition logics in traditional RL tasks like computer games).
>
> The source of confusion, which we will clarify in the paper, is how to model the "randomness" of an external tool. `The key is to recognize that the tool's output is an observation from the environment, not part of the agent's state-transition function`.
>
> Let's use the simple analogy of a dice roll:
>
> - **Stochastic Environment**: The environment rolls a die, and the outcome is 4. This event is stochastic.
> - **Observation**: The agent observes the outcome. This outcome is now a fact.
> - **State Update**: The agent incorporates this fact into its state. For an LLM, this means the token sequence "The dice roll is 4." is now part of the context. Let's call this state $s_t$.
> - **Agent Action**: The agent decides its next action, for example, which is to generate the token "Therefore".
> - **State Transition (Deterministic)**: The transition from $s_t$ to $s_{t+1}$ ($s_t$ + "Therefore") is perfectly deterministic.
>
> The search engine is perfectly analogous to the dice roll. The API call is a query to the environment. The search result is a stochastic observation from that environment. Once the LLM receives this result, these tokens are appended to the history and become part of the new state $s_t$. From this now-fixed state $s_t$, the agent's next action (generating the next token) leads to the next state $s_{t+1}$ with a transition probability of 1.
>
> Therefore, our claim holds: even with stochastic tools (stochasticity in $\mathcal{R}$), the token-level MDP has deterministic transitions (deterministic $\mathcal{P}$) -- which ROVER relies on, and our algorithm (ROVER) that exploit this structure are applicable and can outperform baselines. We will update Appendix I to include this clearer, analogy-based explanation.
>
> ```diff
>   Algorithm 1: LLM Response Rollout with Search Engine Calls (from Search-R1 [1])
>   ...
>   8:          if y_t in [</search>, </answer>, <eos>] then break
>   9:      end while
> + 10:     y ← y + y_b
> + 11:     if <search>...</search> detected in y_b then
> + 12:         Extract search query q ← Parse(y_b, <search>, </search>)
> + 13:         Retrieve search results d = R(q)
> + 14:         Insert d into rollout y ← y + <information>d</information>
>   15:     else if <answer>...</answer> detected in y_b then
>  ...
> ```
> **Reference**
>
> [1] Jin, Bowen, et al. "Search-r1: Training llms to reason and leverage search engines with reinforcement learning." arXiv preprint arXiv:2503.09516 (2025).

---

### Official Review · Reviewer_okcT · 2025-10-30

**Soundness:** 2
**Presentation:** 3
**Contribution:** 2
**Rating:** 2
**Confidence:** 3

**Summary:**

This paper finds that in LLM math reasoning, the optimal action can be recovered from the Q-function of a fixed uniformly random policy, thereby bypassing the generalized policy iteration loop and its associated heuristics. Based on this observation, the authors introduce an algorithm called Random Policy Valuation for Diverse Reasoning (ROVER).

**Strengths:**

1. The paper is clearly written and well motivated.
2. The experiments validate the effectiveness of the method.
3. The method is simple and seems to be scalable.

**Weaknesses:**

1. The motivation and the method itself are very similar to [1]. Although the authors cited [1] in the paper, more detailed discussions are needed, given the similarities. Specifically, [1] provided theoretical results, while the authors stated that "these observations have
remained primarily empirical, with limited theoretical justification." Besides, the authors stated that the main difference lies in that their paper provides the first theoretical analysis that accounts for LLM math reasoning. What are the exact differences that prevent their method and analysis from being useful in math reasoning?
2. Regarding the experiment results, the performance of the baselines is too low in my opinion. For example, in [2], they report 33.33 and 25.42 for AIME 24 and 25, fine-tuned from Qwen3-8b-base using DAPO. The results are 10+ points higher than the same baseline reported in this paper, even outperforming the proposed method by a considerable margin. I understand that the training dataset and hyperparameters are different. But given that (1) their training dataset is barely MATH, instead of DeepScaler which is more in-distribution of AIME, and (2) the performance gap is huge, I believe more careful hyperparameter tuning is needed for both the baselines and the proposed method for a fair comparison.
3. The proposed method does not use random policies. It rather uses the old policy for the rollouts. Thus, the naming is not that appropriate in my opinion. Besides, the main experiments are conducted on Qwen models, which fail to fully demonstrate the effectiveness of the method. From the appendix, fine-tuning on the DeepSeek-R1-Distill-Qwen models seems to give small improvements.

[1] Laidlaw et al., Bridging Reinforcement Learning Theory and Practice with the Effective Horizon.\
[2] Wang et al., Beyond the 80/20 Rule: High-Entropy Minority Tokens Drive Effective Reinforcement Learning for LLM Reasoning.

**Questions:**

See weakness.

---

> ### Author Response · Authors · 2025-11-21
> **Response to Reviewer okcT (Part 1)**
>
> We thank the reviewer for the detailed suggestions and feedback on our work. Please find our responses to each of the concerns below.
>
> > W1: The motivation and the method itself are very similar to [1]. Although the authors cited [1] in the paper, more detailed discussions are needed, given the similarities. Specifically, [1] provided theoretical results, while the authors stated that "these observations have remained primarily empirical, with limited theoretical justification." Besides, the authors stated that the main difference lies in that their paper provides the first theoretical analysis that accounts for LLM math reasoning. What are the exact differences that prevent their method and analysis from being useful in math reasoning?
>
> We appreciate the reviewer highlighting the connection to [1], which we have cited [1] as a valuable stepping stone. However, there are fundamental differences in score, theoretical contributions, and practical applications that merit clarification.
>
> **(Summary of [1])** [1] makes several important contributions to RL theory and practice:
> - Introduces the large-scale BRIDGE dataset (consisting of 155 deterministic MDPs from classic RL benchmarks like Atari, Procgen, and Minigrid)
> - Based on the BRIDGE dataset, **empirically discovers that in approximately 2/3 of these games**, the property that acting greedily with respect to Q-values from a random policy leads to optimal policies holds, and introduces the GORP algorithm that estimates $Q^{\pi_u}$ via random Monte-Carlo rollouts
> - Defines the k-QVI-solvable concept: MDPs are called k-QVI-solvable **if** the above property holds [*Definition 5.1*] (which **treats the property as an assumption (validated empirically)**)
> - Introduces a new theoretical *measure* (effective horizon $H$) that measures *the MDP's complexity* [*Definition 5.2*]
> - Based on the effective horizon measure,
>     - Theoretically analyzes *sample complexity* of GORP [*Lemma 5.3*]
>     - Establishes *bounds on the effective horizon* [*Theorem 5.4*], which provides better sample complexity bounds in the literature
> - Conducts extensive experiments in the BRIDGE dataset consisting of multiple games by comparing with PPO and DQN, considering sample complexity bounds and performance
>
> **(Clarification on theoretical justification)** While [1] provides valuable theoretical results, the specific theoretical justification we refer to as "limited" is the explanation of why certain problems possess the k-QVI-solvable property in the first place. [1] empirically observes this property in approximately 2/3 of tested game environments **but does not theoretically establish why these environments have this property**. Their theoretical contributions focus on defining the k-QVI-solvable concept (assuming it exists), analyzing algorithm complexity given this assumption, and bounding the effective horizon, but **do not involve theoretical proof about what kind of reasoning tasks possess this property**. This is the **"limited theoretical justification" we refer to - not questioning [1]'s theoretical contributions, but highlighting a specific gap our work addresses.**
>
> **(Fundamental theoretical difference)** Our theoretical contributions diverge in crucial ways:
> - **(Proving existence of property: theoretical proof vs. empirical observation)** We theoretically prove that LLM reasoning with verifiable rewards inherently possesses the solvability property (Theorem 1), while [1] empirically validates this property in game environments without theoretical explanation for its existence (but derives sample complexity for MDPs in the BRIDGE dataset consisting of games, while we focus on LLM reasoning)
> - **(New theoretical guarantees)** We establish theoretical bounds for soft sampling approaches (Theorem 2), that preserve diversity while maintaining performance, an aspect not addressed in [1]

---

> > ### Author Response · Authors · 2025-11-21
> > **Response to Reviewer okcT (Part 2)**
> >
> > **(Algorithmic innovations for LLM reasoning)** We focus on LLM reasoning tasks with verifiable rewards in our paper, which present unique challenges compared to classic RL environments studied in [1]. Given these domain differences and challenges, we develop `several key innovations` beyond [1]:
> > - **(Scalability to large action spaces and long horizons)** [1] is based on random Monte-Carlo rollouts, which can become computationally infeasible with LLM reasoning's $T^2A^H$ complexity (exponential in large action space $A$ and long effective horizon $H$). Our approach uses Bellman updates with temporal difference learning (Algorithm 1, line 8) that learns more efficiently [3]. We also provide a practical implementation framework that efficiently scales to LLM math reasoning tasks.
> > - **(Diversity preservation)** We identify and address the "mode collapse" problem in LLM reasoning (Section 3.1.1). Our soft sampling approach maintains diversity in reasoning paths while still providing theoretical performance guarantees. This diversity-preserving mechanism is crucial for LLM reasoning but not studied in [1]'s domains (in games, the more important part is quality instead of diversity [5]).
> >
> > **(Significance of Our Results)** In traditional RL, it is generally believed that random policies provide insufficient guidance toward optimal policies in complex tasks. While [1] empirically discovered this property in some game environments, our work **provides the first theoretical explanation for why this property specifically exists in LLM reasoning tasks**. By proving this property theoretically rather than observing it empirically, we establish a fundamental understanding of LLM reasoning that enables principled algorithm design with performance guarantees. Our contributions not only build upon [1]'s important empirical observations but extend them with rigorous theoretical foundations specific to LLM reasoning, along with practical algorithms that address the unique challenges of this domain with novel designs and effective performance improvement and diversity preservation.
> >
> > > W2: Regarding the experiment results, the performance of the baselines is too low in my opinion. For example, in [2], they report 33.33 and 25.42 for AIME 24 and 25, fine-tuned from Qwen3-8b-base using DAPO. The results are 10+ points higher than the same baseline reported in this paper, even outperforming the proposed method by a considerable margin. I understand that the training dataset and hyperparameters are different. But given that (1) their training dataset is barely MATH, instead of DeepScaler which is more in-distribution of AIME, and (2) the performance gap is huge, I believe more careful hyperparameter tuning is needed for both the baselines and the proposed method for a fair comparison.
> >
> > Thanks for the question. We have conducted additional new experiments following [2] to address the concern, and please also note the fair comparison in our previous experiments (following established practices with different datasets and max response length, as explained below).
> >
> > Upon the recent code release from [2] (after paper submission), we conduct additional experiments following their dataset composition and setup using the open-source codebase from [2] (https://github.com/Shenzhi-Wang/Beyond-the-80-20-Rule-RLVR). The new results on AIME2024 are presented below, demonstrating that **ROVER consistently outperforms the baseline under this configuration even without careful hyperparameter tuning**, which achieves a 2.3% improvement over the DAPO baseline. Notably, ROVER generates significantly longer average response length (10,464 tokens compared to DAPO's 6,884 tokens). **ROVER also excels in diversity, showing a 15% relative increase in the number of distinct strategies and a better utility** (the diversity metric used in our paper, proposed by NoveltyBench*, confirming our paper's core claim about maintaining both quality and diversity. We will release our newly trained ROVER model on Hugging Face.
> >
> > |   Method  | avg@16    | len@16       | # of distinct strategies | Utility   |
> > | ----- | --------- | ------------ | ------------------------ | --------- |
> > | DAPO  | 33.33     | 6884.89      | 1.92                    | 0.223     |
> > | ROVER | **35.63** | **10464.43** | **2.21**                | **0.245** |

---

> > > ### Author Response · Authors · 2025-11-21
> > > **Response to Reviewer okcT (Part 3)**
> > >
> > > We would like to clarify that our original experimental setup followed standard practices and the widely-adopted configuration established in prior work [6], and **ensured fair comparisons across all methods**. We summarize the key differences between our dataset and setup and [2] as follows:
> > >
> > > | Setting                 | Our Manuscript | Beyond 80/20 Rule [2] |
> > > | :-- | :-- | :- |
> > > | Maximum Response Length | 8,192          | 20,480                |
> > > | Training Dataset        | DeepScaler     | DAPO-Math-17k         |
> > >
> > > The 8K maximum response length used in our manuscript means that during RL training, the model's responses are truncated more frequently than with a 20k maximum response length. **We believe that the shorter maximum response length (8K vs 20K) is the primary factor for the relatively lower performance of ROVER and baseline models** in our previous experiments (as the model may not receive effective reward signals due to truncation).
> > >
> > > `To ensure full transparency and reproducibility, we will open-source both our codebase and all trained models under this added experiment`.
> > >
> > > **In summary**, these new results conclusively demonstrate that ROVER's core advantage: achieving strong performance with a significantly simpler algorithm, holds across different experimental configurations. The simplicity-to-performance ratio remains ROVER's primary contribution, as it eliminates complex heuristics while matching or exceeding sophisticated methods.
> > >
> > > > W3.1: The proposed method does not use random policies. It rather uses the old policy for the rollouts. Thus, the naming is not that appropriate in my opinion.
> > >
> > > Thanks for the question. We understand the potential confusion and would like to clarify that we `indeed estimate the value for the random policy`. Please note that there is a **fundamental distinction** between the rollout policy (used to generate trajectories) and the policy being evaluated. In our method, we explicitly evaluate the value function of a uniform random policy $\pi_u$ through our updates (line 8 in Algorithm 1).
> > >
> > > According to the Bellman equation, the Q-function for $\pi_u$ is defined as $Q^{\pi\_u}(s, a) = R(s, a) + \gamma \mathbb{E}\_{s' \sim P(\cdot|s,a)} [ V^{\pi\_u}(s')]$, where $V^{\pi\_u}(s') = \mathbb{E}\_{a' \sim \pi\_u(\cdot|s')} \left[ Q^{\pi\_u}(s', a') \right]
> > >  = \frac{1}{|A(s')|} \sum\_{a' \in A(s')} Q^{\pi\_u}(s', a')$. Therefore, $Q^{\pi_u}(s,a) = R(s, a) + \gamma  \sum_{s'} P(s'|s,a) \left[ \frac{1}{|A(s')|} \sum_{a' \in A(s')} Q^{\pi_u}(s', a') \right]$.
> > >
> > > In the context of math reasoning (which has deterministic transition dynamics), and the discount factor $\gamma$ is set to $1$, this simplifies to $Q^{\pi_u}(s,a) = R(s,a) + \frac{1}{|A(s')|} \sum_{a' \in A(s')} Q^{\pi_{u}}(s', a')$.
> > >
> > > It is important to note that by using the mean operator [3,4,5], i.e., ${\text{mean}}(Q^{\pi_u}(s',\cdot))=\frac{1}{|A(s')|} \sum_{a' \in A(s')} Q^{\pi_{u}}(s', a')$, we can evaluate the Q-function for the random policy. Our method exactly implements this mean operator for value backup, which directly corresponds to evaluating the random policy's value function -- regardless of which policy generated the rollout trajectories (with theoretical guarantees presented in [4] for using alternative operators for performing value updates).
> > >
> > > Therefore, "random policy valuation" accurately describes the core theoretical foundation of our approach, even though we leverage a more efficient policy for trajectory sampling, considering the large state and action spaces.
> > >
> > > > W3.2: Besides, the main experiments are conducted on Qwen models, which fail to fully demonstrate the effectiveness of the method. From the appendix, fine-tuning on the DeepSeek-R1-Distill-Qwen models seems to give small improvements.
> > >
> > > Thanks for the point about model diversity. We have conducted additional experiments using LLaMA-based models, which represent a distinct model family from Qwen.
> > >
> > > **(Results on a different model family)** We trained the `Llama3.1-8B-Instruct` model on the hard training dataset from simpleRL-zoo [7], which is composed of level 3-5 prompts from the MATH training set. As shown in the table below (included in the updated manuscript in Appendix H), **ROVER achieves an avg@5 accuracy score of `67.8%` on MATH500, an improvement of `8.8%` over the model trained with GRPO.** This demonstrates the broad applicability of ROVER to different model architectures.
> > >
> > > | MATH500                     |  avg@5 |
> > > | :------------------- | :------------ |
> > > | Llama3.1-8B-Instruct + ROVER               | **67.8**      |
> > > | Llama3.1-8B-Instruct + GRPO                | 59.0          |
> > > | Llama3.1-8B-Instruct | 51.9          |

---

> > > > ### Author Response · Authors · 2025-11-21
> > > > **Response to Reviewer okcT (Part 4)**
> > > >
> > > > **(Regarding the DeepSeek-R1-Distill-Qwen improvements)** As detailed in Appendix F, the DeepSeek-R1-Distill-Qwen-1.5B model was trained for only approximately 120 hours on 8 GPUs. Despite this limited post-training computational budget, ROVER yielded substantial improvements over the base model, increasing accuracy from 29.3% to 42.2% on AIME24 and 24.3% to 31.2% on AIME25. We emphasize that these gains are significant, particularly given ROVER's superior **pass@64** performance compared to both ProRL and DeepScaleR. Crucially, ROVER improves pass@1 accuracy without compromising the pass@k score, demonstrating its unique advantage in **preserving reasoning diversity**. Furthermore, in the context of 1.5B-parameter models, this magnitude of improvement is widely recognized in the literature as a substantial advancement [8,9].
> > > >
> > > > **Reference**
> > > >
> > > > [1] Laidlaw et al., Bridging Reinforcement Learning Theory and Practice with the Effective Horizon.
> > > >
> > > > [2] Wang et al., Beyond the 80/20 Rule: High-Entropy Minority Tokens Drive Effective Reinforcement Learning for LLM Reasoning.
> > > >
> > > >
> > > > [3] Littman, Michael Lederman. Algorithms for sequential decision-making. Brown University, 1996.
> > > >
> > > >
> > > > [4] Asadi, Kavosh, and Michael L. Littman. "An alternative softmax operator for reinforcement learning." International Conference on Machine Learning. PMLR, 2017.
> > > >
> > > > [5] He, Haoran, et al., Random Policy Evaluation Uncovers Policies of Generative Flow Networks, ICML 2025
> > > >
> > > > [6] Luo, Michael, et al. "DeepScaleR: Surpassing O1-Preview with a 1.5B Model by Scaling RL." Notion, 2025, pretty-radio-b75.notion.site/DeepScaleR-Surpassing-O1-Preview-with-a-1-5B-Model-by-Scaling-RL-19681902c1468005bed8ca303013a4e2.
> > > >
> > > > [7] Zeng, Weihao, et al. "Simplerl-zoo: Investigating and taming zero reinforcement learning for open base models in the wild." arXiv preprint arXiv:2503.18892 (2025).
> > > >
> > > > [8] Liu, Runze, et al. "Attention as a Compass: Efficient Exploration for Process-Supervised RL in Reasoning Models." Arxiv 2025
> > > >
> > > > [9] Chen, Xiwen, et al. "Dra-grpo: Exploring diversity-aware reward adjustment for r1-zero-like training of large language models." arXiv 2025.
> > > >
> > > > ---
> > > >
> > > > **We thank the reviewer for the time and effort in reviewing our work! We would very much appreciate it if the reviewer could check our responses and the updates in the paper and let us know whether they address your concerns. If they have, we would appreciate it if you could kindly consider raising the score. If not, we are always willing to address any of your further concerns.**

---

> ### Author Response · Authors · 2025-11-26
> **Looking forward to your reply!**
>
> Dear Reviewer okcT,
>
> We hope this message finds you well. We are writing to kindly follow up on our response to your valuable comments and questions, which we submitted over 5 days ago.
>
> In our rebuttal,  we addressed your comments by **clarifying the novel contributions of our methodology and results that go beyond the scope of  [1], incorporating new experiments following the setting of [2], and adding further validation on the Llama-3.1-8B-Instruct model**.
>
> Your feedback has been crucial in refining our work, and we greatly appreciate the time and effort you have invested in reviewing our paper. If you have any additional thoughts or questions about our responses, we would be more than happy to address them.
>
> Thank you once again for your insightful review! We look forward to your further feedback.
>
> Best,
>
> The Authors
>
> [1] Laidlaw et al., Bridging Reinforcement Learning Theory and Practice with the Effective Horizon.
>
> [2] Wang et al., Beyond the 80/20 Rule: High-Entropy Minority Tokens Drive Effective Reinforcement Learning for LLM Reasoning.

---

> > ### Comment · Reviewer_okcT · 2025-11-27
> >
> > Thanks to the authors for the detailed reply. I decide to increase the score.

---

> > > ### Author Response · Authors · 2025-11-28
> > >
> > > Thanks a lot for your feedback and for reconsidering our work! We are glad that our response has addressed the concerns, and we greatly appreciate the time and effort the reviewer has dedicated to reviewing our paper.

---

### Official Review · Reviewer_8GQr · 2025-10-31

**Soundness:** 3
**Presentation:** 3
**Contribution:** 3
**Rating:** 6
**Confidence:** 4

**Summary:**

This paper simplifies reinforcement learning with verifiable rewards (RLVR) for LLM reasoning. Exploiting the deterministic, tree-structured MDP with binary rewards, the authors prove that evaluating a uniform random policy's Q-function suffices for optimal reasoning—bypassing iterative policy improvement (PPO/GRPO). The proposed ROVER algorithm samples via softmax over these Q-values, balancing diversity and performance. Experiments show significant gains in reasoning quality and diversity versus complex RL baselines.

**Strengths:**

1. Algorithmic Minimalism: ROVER dispenses with conventional policy improvement loops and ad hoc stabilizers (e.g., KL penalties or clipping), yielding a strikingly simple and easy-to-tune algorithm, as clearly depicted in Figure 3 (p. 3).

2. Strong Empirical Performance and Strategic Diversity: Across standard math reasoning and countdown tasks, ROVER consistently outperforms strong baselines (e.g., GRPO, DAPO, REINFORCE++) in both solution accuracy (Table 1, p. 8; Figure 9) and the generation of diverse, correct reasoning paths (Figures 7, 10, 17).

3. Reproducibility: Public release of source code enhances transparency and facilitates replication.

**Weaknesses:**

1. Narrow Theoretical Scope: The optimality guarantee for ROVER strictly requires deterministic, tree-structured MDPs with binary terminal rewards. The paper offers no empirical or theoretical exploration of how—or whether—the method degrades gracefully under violations of these assumptions (e.g., stochasticity, intermediate rewards, or graph-structured environments), limiting its applicability to more realistic RLVR settings.

2. Oversimplified Reward Modeling: ROVER assumes only terminal binary feedback, ignoring process-based signals, partial credit, or intermediate rewards increasingly used in LLM alignment. The paper provides no discussion of how the framework might extend to richer reward structures, hindering practical deployment.

3. Limited Model Scale in Experiments: All evaluations use models up to 8B parameters, falling short of current state-of-the-art scales (e.g., 70B+). This constraint tempers claims about ROVER’s readiness for deployment in cutting-edge LLM systems.

**Questions:**

1. Scaling to Longer Horizons and Larger Action Spaces. How does ROVER’s computational and memory footprint compare to PPO/GRPO at the largest vocabularies and problem sizes considered? Can the authors clarify if action batching, logit pruning, or any approximations are required to keep training feasible?

2. Reward Shaping/Process-level Signals. Have the authors tested ROVER with partial or intermediate rewards (e.g., for partially correct math steps or process rewards)? If so, what is observed? If not, are there theoretical or practical obstacles?

3. Extension to Non-binary, Non-deterministic Environments. How does ROVER behave when rewards are non-binary or transitions are stochastic? Is there a degradation in quality/diversity, and are there principled ways to extend the theorem to these cases?

4. Failure and Negative Cases. Can the authors discuss cases where ROVER’s performance flatlines or drops compared to PPO/GRPO (e.g., pathological tasks, extremely long horizons, or ambiguous reward settings)?

---

> ### Author Response · Authors · 2025-11-21
> **Response to Reviewer 8GQr (Part 1)**
>
> We thank the reviewer for the careful reading and valuable feedback on our work, and we greatly appreciate the comments!
>
> > W1: Narrow Theoretical Scope: The optimality guarantee for ROVER strictly requires deterministic, tree-structured MDPs with binary terminal rewards. The paper offers no empirical or theoretical exploration of how—or whether—the method degrades gracefully under violations of these assumptions (e.g., stochasticity, intermediate rewards, or graph-structured environments), limiting its applicability to more realistic RLVR settings.
>
> Thanks for the question. We would like to clarify why our framework has broader application than it might initially appear, and our additional analysis addresses both theoretical extensions and empirical evidence demonstrating its broad applicability:
>
> ### Theoretical extensions & Empirical evidence of robustness
>
> **(Theoretical extensions)** Although current benchmarks typically use outcome-based terminal rewards, we have also extended our theoretical results to accommodate intermediate feedback (we included the proof and detailed discussion in Appendix A.3):
>
> `Theorem 3`: Consider a finite-horizon MDP with deterministic transitions and a tree-structured state space. Let $\pi_u$ be a uniform policy, $Q^{\pi_u}$ denote its Q-function, and the discount factor $\gamma=1$. For any non-terminal state $s$, define $\mathcal{C}$ to be the set of terminal states with correct rewards, $T(s,a)$ the next state reached by taking action $a$ from state $s$, and $\mathcal{T}(s)$ the set of all possible trajectories from state $s$ to terminal states. Define two disjoint subsets of actions at state $s$, where $\mathcal{A}^+(s)=\{a \in \mathcal{A} | \exists \tau \in \mathcal{T}(T(s,a))\ {\rm such\ that}\ \tau\ {\rm ends\ in}\  \mathcal{C}\}$, and $\mathcal{A}^-(s)=\{a \in \mathcal{A} | \forall \tau \in \mathcal{T}(T(s,a)), \tau\ {\rm does\ not\ ends\ in}\ \mathcal{C}\}$. The intermediate rewards are given by a reasonable Process Reward Model (PRM) $r$. Terminal rewards $R$ indicate a correct solution ($R > 0$), and $0$ otherwise. A PRM is considered reasonable if the following conditions hold for all non-terminal states $s$: (i) $\forall a_{+}^{'} \in \mathcal{A}^+(s), \ \forall a_{-}^{'} \in \mathcal{A}^-(s)$: $r(s, a^-) < r(s, a^+)$, and (ii) $\forall a_{+} \in \mathcal{A}^+(s), \ \forall a_{-} \in \mathcal{A}^-(s), \ \forall \tau^- \in \mathcal{T}(T(s,a^-)), \ \forall \tau^+ \in \mathcal{T}(T(s,a^+))$: $\sum_{i \in \tau^-} r_i < \sum_{i \in \tau^+} r_i$. If these reasonable PRM conditions hold, then the greedy policy with respect to $Q^{\pi_u}$ defined by $\pi_{\rm greedy}(s) = \arg\max_a Q^{\pi_u}(s,a)$ always selects actions leading to a correct terminal state.
>
> This extension demonstrates that our novel theoretical framework and algorithm remain effective, as intermediate rewards might be provided by process reward models throughout the trajectory (under the reasonable assumption -- please note that if the process reward model is not reasonable, it will also affect PPO/GRPO's learning due to their reward-maximizing behavior).

---

> > ### Author Response · Authors · 2025-11-21
> > **Response to Reviewer 8GQr (Part 2)**
> >
> > **(Empirical evidence of robustness)** Current LLM reasoning RLVR tasks inherently exhibit deterministic dynamics (even tool-use scenarios maintain deterministic token-level dynamics as detailed below) and tree structures (from autoregressive generation as detailed below). Nevertheless, to thoroughly investigate ROVER's performance beyond its theoretical guarantees under these natural properties, we conduct additional experiments designed to **evaluate our method's robustness on environments with intermediate rewards, stochastic transitions, and graph-structured MDPs**:
> > 1. **Intermediate rewards**: Since the RLVR problems basically follow the deterministic dynamic transitions with a terminal reward that indicates the correctness, we modify the didactic MDP in Figure 5 in our paper to validate the practical effectiveness of ROVER for intermediate rewards and stochastic transitions. Concretely, we assign a reward of 1 to the intermediate states {$\{A,B,C,D,AC,BD,CA,DB\}$} that can lead to the correct terminal state. In this environment, ROVER still captures all optimal solutions with a 100\% success rate. (Details provided in Appendix C.1)
> > 2. For **stochastic transitions**, we follow the widely-adopted sticky action concept [5] in classic RL literature to inject stochasticity in token-level transition dynamics. Specifically, we modify the environment dynamics to be stochastic by setting $P(s+a|s,a)=0.75, P(s+a'|s,a)=0.25$, where $a'$ denotes the action different from the chosen action $a$. This means that there is a 0.25 probability for the agent to travel a random path, thereby ensuring the dynamic stochastic. Our empirical results demonstrate that ROVER can effectively handle this kind of stochasticity. If the dynamic becomes deterministic at test time, ROVER successfully captures all modes with a 100\% success rate. If the dynamic remains stochastic at test time, ROVER achieves a 77\% success rate, which is still high considering the 25\% randomness in the environment. (detailed performance discussions are included in Appendix C.2).
> > 3. **Graph-structured MDPs**: Similarly, as standard RLVR tasks follow the tree-structured MDP strictly, we evaluate the performance of ROVER on graph structure using the didactic example provided in Figure 5 in the main text. We insert a graph connection between {$\{A,B,C,D\}$}, which means a state can have multiple parent nodes, leading to a graph-structured MDP. We provide an illustration of this graph-structured MDP in Appendix C.3 in our updated manuscript. Our empirical results show that ROVER is still optimal under graph-structured MDPs, achieving a 100\% success rate with all four modes covered. (Details are provided in Appendix C.3)
> >
> > ### Additional experiments on Tool-Use
> > **We conduct additional experiments to validate ROVER in tool-use scenarios based on the open-sourced Search-R1 [1]'s setup**. Specifically, we fine-tune the Qwen2.5-3B-Base model on the combined training sets of Natural Questions (NQ) and HotpotQA. Given limited time and resources during the rebuttal phase, we evaluate ROVER trained with only 300 steps (lower than 500 steps of GRPO in the original Search-R1 paper [1]). However, ROVER still outperforms the baseline by an obvious margin, which demonstrates its great potential in tool-use experiments. Note that we did not tune any hyperparameters in ROVER in this experiment.
> > **(Performance comparison)** The performance across multiple benchmarks, spanning both General QA and Multi-hop QA, is presented in the table below, which **shows that ROVER not only works effectively with tools but actually outperforms the original GRPO in terms of average performance (even without extensive hyperparameter tuning of ROVER)**:
> >
> > |                       | NQ        | TriviaQA  | PopQA     | HotpotQA | 2wiki     | Musique   | Avg.      |
> > | --------------------- | --------- | --------- | --------- | -------- | --------- | --------- | --------- |
> > | Search-R1-base (GRPO) | 0.421     | 0.583     | 0.413     | 0.297    | 0.274     | 0.066     | 0.342     |
> > | ROVER                 | **0.477** | **0.600** | **0.443** | 0.290    | **0.286** | **0.069** | **0.360** |
> >
> >
> > **(Diversity comparison)** To evaluate diversity, we generated 32 responses for each prompt from the Bamboogle benchmark (with the temperature set to 1.0). We then use the cosine distance between agent responses as the diversity metric. The results in the table below indicate that ROVER shows better diversity.
> >
> > |     Method      | cosine distance (evaluated on Bamboogle) |
> > | :-------------------- | :----------------------------------------------------- |
> > | Search-R1-base (GRPO) | 0.042                                                 |
> > | ROVER                 | **0.065**                                                 |

---

> > > ### Author Response · Authors · 2025-11-21
> > > **Response to Reviewer 8GQr (Part 3)**
> > >
> > > We argue that diversity is crucial for tool-use tasks, which also represents a potential future direction for our simple yet effective ROVER algorithm. An agent capable of generating diverse queries can retrieve more varied and comprehensive information from the corpus, thereby increasing its chances of discovering the key evidence required to correctly answer a prompt. Conversely, a model lacking diversity may fall into monotonous query patterns, leading to a failure loop where it repeatedly fetches the same unhelpful results from the search engine and is unable to solve the problem.
> > >
> > > In the following part, we explain why ROVER works well even with complicated tool uses.
> > >
> > > ### Natural properties of LLM reasoning
> > >
> > > The key MDP properties (deterministic transition dynamics and tree structure) in our theory are not limiting assumptions but rather reflect the inherent nature of realistic RLVR settings, while we have also extended our theory to incorporate intermediate feedback to further broaden its scope.
> > >
> > > 1. **Transition dynamics**: This property naturally emerges from token-level action definition in LLM reasoning in RLVR tasks. **Even with tools that appear stochastic (e.g., search engines), the MDP with token-level action definition still has deterministic transition dynamics.** Consider a search tool example in Search-R1 [1] (or similar tool-use setups): when an LLM queries a search engine within its reasoning process, the search results may appear "random" or unpredictable. However, based on the MDP formulation with token-level action, the search result (whether successful or failed) becomes part of the state once received, and the next token generation is deterministic conditioned on this state.
> > >
> > > **Example**: Given a query "... \</search>", if the search engine responds "This is 1", then:
> > > P(next_state="... \</search>\<information> This" | state="... \</search>\<information>", action="This")=1. Similarly, for a failed search (with response "Fail"):
> > > P(next_state="... \</search>\<information> Fail" | state="... \</search><\information>", action="Fail")=1
> > >
> > >  The key insight is that in the MDP (with token-level action definition), the agent's actions are individual tokens it generates, not the API calls themselves. The source of uncertainty (external tool) becomes fixed information in the state (analogous to the random outcomes and deterministic mappings in random variables). This applies to various LLM reasoning scenarios with verifiable rewards (e.g., Search-R1 [1]).
> > >
> > > 2. **Tree structure**: This directly follows from autoregressive language generation, where each token choice (action) creates a new branch (next state) in the reasoning path, and the sequence can never cycle back to a previous state (unlike navigation or control problems that may lead to graphs or even cyclic graphs -- posing significant challenges for value function estimation), which holds universally across all reasoning domains based on autoregressive generation.
> > >
> > > 3. **Reward structure**: We follow current popular practices in reasoning tasks with verifiable rewards that are based on outcome rewards [2,3,4,5], as the correctness is evaluated upon completion, and this kind of reward signal is already effective to guide the agent's RL training.
> > >
> > > Rather than working in simplified settings or considering limited assumptions, we have identified the actual structural properties of RLVR tasks based on LLMs that enable more efficient algorithms.
> > >
> > > **In summary, our theoretical extensions and additional experiments show that ROVER maintains strong performance in terms of solution quality and diversity advantages across a range of conditions.** These results not only validate ROVER's robustness but also suggest that our fundamental insights about leveraging the inherent structure of reasoning tasks could inspire new research directions beyond current RLVR paradigms, potentially leading to a new class of RL algorithms that more efficiently exploit the natural properties of language model reasoning.

---

> ### Author Response · Authors · 2025-11-21
> **Response to Reviewer 8GQr (Part 4)**
>
> > W2: Oversimplified Reward Modeling: ROVER assumes only terminal binary feedback, ignoring process-based signals, partial credit, or intermediate rewards increasingly used in LLM alignment. The paper provides no discussion of how the framework might extend to richer reward structures, hindering practical deployment.
>
> Thanks for the question about reward modeling. While many current RLVR algorithms are trained based on binary terminal rewards [2,3,4], we agree that richer reward structures are also critical for alignment. To address this concern, we have extended ROVER's theoretical framework to incorporate intermediate rewards. Our additional theoretical analysis (Theorem 3 in Appendix A.3) proves that ROVER maintains optimality guarantees given reasonable process reward models that provide intermediate feedback throughout the reasoning trajectory, as detailed in the response to W1 (specifically, when the PRM assigns higher rewards to prefixes of correct trajectories than to incorrect ones). We also conducted additional evaluations of ROVER when given intermediate rewards, which is also detailed in the response to W1.
>
> In summary, **our theoretical extensions and experiments demonstrate that ROVER effectively accommodates process reward models and maintains strong performance advantages in both solution quality and diversity**, making it practical for deployment.
>
> > W3: Limited Model Scale in Experiments: All evaluations use models up to 8B parameters, falling short of current state-of-the-art scales (e.g., 70B+). This constraint tempers claims about ROVER’s readiness for deployment in cutting-edge LLM systems.
>
> Due to computational constraints, we did not test our model with 70B+ parameters. Our results strongly support ROVER's applicability across scales and with different model families (additional experiments based on Llama in Appendix H) in a number of benchmark tasks, with consistent performance and diversity advantages over baselines as model size grows. We are actively pursuing validation on larger models (e.g., 70B+ as suggested) as resources permit.
>
> > Q1.1: Scaling to Longer Horizons and Larger Action Spaces. How does ROVER’s computational and memory footprint compare to PPO/GRPO at the largest vocabularies and problem sizes considered?
>
>
> We conduct extensive experiments in widely-adopted standard benchmarks, and please note that vocabulary size (action space) remains the same across diverse reasoning tasks. Regarding the horizon, we followed standard setups for all comparisons, and we have also added an additional experiment testing scaling to longer context lengths to 20k, where ROVER still outperforms strong baselines.
>
> ROVER exhibits substantial efficiency advantages. ROVER's memory advantage stems from its simpler architecture requirements (for example, when using Qwen3-8B-Base as the base model):
> - PPO requires `three` models, including the critic model, the policy model, and the reference policy model (`24B parameters in total`)
> - GRPO requires `two` models, including the policy model and the reference policy model (`16B parameters in total`)
> - ROVER requires only `a single` policy model (`8B parameters`)
>
> Since ROVER requires minimal memory usage for models, it has more memory for the KV cache and thus has increased training speed.
>
> > Q1.2: Can the authors clarify if action batching, logit pruning, or any approximations are required to keep training feasible?
>
> We do not assume these approximations throughout all of the experiments in our paper, where we follow standard practices as in [5], and all implementations are discussed in the main text and appendix, with open-sourced code and model.
>
> > Q2: Reward Shaping/Process-level Signals. Have the authors tested ROVER with partial or intermediate rewards (e.g., for partially correct math steps or process rewards)? If so, what is observed? If not, are there theoretical or practical obstacles?
>
> Please see our detailed responses to W1 and W2 for our theoretical extensions and additional empirical validation under intermediate rewards (also detailed in Appendix A.3), where ROVER still maintains its robust performance with a theoretical guarantee.
>
> > Q3: Extension to Non-binary, Non-deterministic Environments. How does ROVER behave when rewards are non-binary or transitions are stochastic? Is there a degradation in quality/diversity, and are there principled ways to extend the theorem to these cases?
>
> Please see our detailed responses to W1 and W2, where we extend our theory beyond binary terminal rewards with empirical validations, and our responses to W1 regarding stochastic transition dynamics, where ROVER still maintains its robust performance with a theoretical guarantee.

---

> > ### Author Response · Authors · 2025-11-21
> > **Response to Reviewer 8GQr (Part 5)**
> >
> > > Q4: Failure and Negative Cases. Can the authors discuss cases where ROVER’s performance flatlines or drops compared to PPO/GRPO (e.g., pathological tasks, extremely long horizons, or ambiguous reward settings)?
> >
> > It is important to note that we follow standard setups for LLM reasoning tasks with verifiable rewards (given well-defined rewards rather than pathological tasks or ambiguous rewards). We have conducted additional experiments with even longer CoT lengths (20k), and across all standard benchmarks, ROVER consistently outperforms PPO/GRPO in terms of performance and diversity.
> >
> > During our development, we find that ROVER performs best when initialized with a reasonably good policy model. With poor initialization, learning efficiency may be reduced. This is an interesting direction for future work. Given a lower-quality initial model, it will be valuable to explore further SFT before applying ROVER.
> >
> > **Reference**
> >
> > [1] Jin, Bowen, et al. "Search-r1: Training llms to reason and leverage search engines with reinforcement learning." arXiv preprint arXiv:2503.09516 (2025).
> >
> > [2] Liu, Zihe, et al. "Part i: Tricks or traps? a deep dive into rl for llm reasoning." arXiv preprint arXiv:2508.08221 (2025).
> >
> > [3] Zhang, Kaiyan, et al. "A survey of reinforcement learning for large reasoning models." arXiv 2025.
> >
> > [4] Zeng, Weihao, et al. "Simplerl-zoo: Investigating and taming zero reinforcement learning for open base models in the wild." COLM 2025.
> >
> > [5] Luo et al., DeepScaleR: Surpassing O1-Preview with a 1.5B Model by Scaling RL, Notion Blog 2025
> >
> >
> > ---
> >
> > We thank the reviewer for the time and effort in reviewing our work! We would very much appreciate it if the reviewer can check our responses and the updates in the paper and let us know whether they address your concerns. We are happy to provide further clarification if you have any additional concerns.

---

> ### Author Response · Authors · 2025-11-26
> **Looking forward to your reply!**
>
> Dear Reviewer 8GQr,
>
> We are writing to kindly follow up on our response to your valuable comments and questions, which we submitted over 5 days ago.
>
> In our rebuttal, we provided a new theorem to validate the generalizability of our method to new environments (`see Theorem 3 in Appendix A.3 and new experiments in Appendix C`), additional experiments on the tool-use domain to further validate the superior performance of our method (`see Appendix I`), and detailed clarifications to demonstrate the robustness of our propose method.
>
> Your feedback has been crucial in refining our work, and we greatly appreciate the time and effort you have invested in reviewing our paper. If you have any additional thoughts or questions about our responses, we would be more than happy to address them.
>
> Thank you once again for your insightful review! We look forward to your further feedback.
>
> Best,
>
> The authors

---

### Official Review · Reviewer_Qj2i · 2025-10-31

**Soundness:** 3
**Presentation:** 2
**Contribution:** 3
**Rating:** 6
**Confidence:** 4

**Summary:**

This paper proposes ROVER (Random Policy Valuation for Diverse Reasoning), the key idea is the optimal policy can be derived simply by greedily selecting actions based on the Q-values of a fixed, uniformly random policy. This bypasses the need for complex Generalized Policy Iteration (GPI) loops used in standard algorithms like PPO and GRPO. To maintain diversity (which pure greedy selection sacrifices), ROVER samples actions via a softmax over these uniform-policy Q-values. Empirically, ROVER outperforms strong baselines (GRPO, REINFORCE++, DAPO) on math benchmarks (AIME, MATH500, etc.) in both solution quality.

**Strengths:**

1. The core finding—that a uniform random policy's Q-values are sufficient for optimal control in this specific MDP structure—is a significant and surprising theoretical contribution that challenges prevailing RL paradigms for this domain.
2. ROVER drastically simplifies the RL pipeline by removing actor-critic loops, reference models (explicitly), and complex hyperparameter tuning associated with PPO/GRPO. It is computationally efficient and easy to implement.
3. The method consistently outperforms state-of-the-art baselines across multiple models (Qwen-4B, 8B, DeepSeek-1.5B) and challenging benchmarks (AIME, HMMT), showing gains in both pass@1 and pass@k (diversity).

**Weaknesses:**

1. The theoretical guarantee relies on a strict MDP structure (deterministic, tree-structured, binary terminal rewards). While standard for current math benchmarks, this may not hold for more complex reasoning tasks involving tool use, intermediate feedback, or stochastic environments, potentially limiting broader applicability.
2. While the paper claims robustness, the ablation shows performance is sensitive to the temperature parameter $\rho$. An extremely low $\rho$ causes collapse, while a high $\rho$ reduces to random sampling. The default $\rho=1$ works well here but might need tuning for other domains.

**Questions:**

1. Theorem 1 proves that acting greedily according to the random policy's Q-values is optimal. Why, then, does the practical ROVER algorithm use a softmax sampler instead of a purely greedy approach?
2. How would this be helpful if extend to other more complicated domain than math?

---

> ### Author Response · Authors · 2025-11-21
> **Response to Reviewer Qj2i (Part 1)**
>
> We thank the reviewer for the careful reading and valuable feedback on our work, and we greatly appreciate the comments!
>
> > W1: The theoretical guarantee relies on a strict MDP structure (deterministic, tree-structured, binary terminal rewards). While standard for current math benchmarks, this may not hold for more complex reasoning tasks involving tool use, intermediate feedback, or stochastic environments, potentially limiting broader applicability.
>
> Thanks for the question. We would like to clarify why our framework has a broader application than it might initially appear:
>
> ### Natural properties of LLM reasoning tasks (not strict limitations) & Additional theoretical results
>
> The key MDP properties (deterministic transition dynamics and tree structure) in our theory are not limiting assumptions but rather reflect the inherent nature of general RLVR tasks with LLM reasoning (math, code, tool use as in Search-R1 [1]) with verifiable rewards, while we have also extended our theory to incorporate intermediate feedback to further broaden its scope.
>
> 1. **Transition dynamics**: This property naturally emerges from token-level action definition in LLM reasoning in RLVR tasks. **Even with tools that appear stochastic (e.g., search engines), the MDP with token-level action definition still has deterministic transition dynamics.** Consider a search tool example in Search-R1 [1] (or similar tool-use setups): when an LLM queries a search engine within its reasoning process, the search results may appear "random" or unpredictable. However, based on the MDP formulation with token-level action, the search result (whether successful or failed) becomes part of the state once received, and the next token generation is deterministic conditioned on this state.
>
> **Example**: Given a query "... \</search>", if the search engine responds "This is 1", then:
> P(next_state="... \</search>\<information> This" | state="... \</search>\<information>", action="This")=1. Similarly, for a failed search (with response "Fail"):
> P(next_state="... \</search>\<information> Fail" | state="... \</search><\information>", action="Fail")=1
>
> The key insight is that in the MDP (with token-level action definition), the agent's actions are individual tokens it generates, not the API calls themselves. The source of uncertainty (external tool) becomes fixed information in the state (analogous to the randomness outcomes and deterministic mappings in random variables). This applies to various LLM reasoning scenarios with verifiable rewards (e.g., Search-R1 [1]). We have conducted additional experiments based on Search-R1 [1] as below, which demonstrates ROVER's effectiveness.
>
> 2. **Tree structure**: This directly follows from autoregressive language generation, where each token choice (action) creates a new branch (next state) in the reasoning path, and the sequence can never cycle back to a previous state (unlike navigation or control problems), which holds universally across all reasoning domains based on autoregressive generation.
>
> 3. **Reward structure**: We follow current popular practices in reasoning tasks with verifiable rewards that are based on outcome rewards [2,3,4], as the correctness is evaluated upon completion, and this kind of reward signal is already effective to guide the agent's RL training.
>
> Although current benchmarks typically use outcome-based terminal rewards, we have extended our theoretical results to accommodate intermediate feedback (we included the proof and detailed discussion in Appendix A.3):

---

> ### Author Response · Authors · 2025-11-21
> **Response to Reviewer Qj2i (Part 2)**
>
> `Theorem 3`: Consider a finite-horizon MDP with deterministic transitions and a tree-structured state space. Let $\pi_u$ be a uniform policy, $Q^{\pi_u}$ denote its Q-function, and the discount factor $\gamma=1$. For any non-terminal state $s$, define $\mathcal{C}$ to be the set of terminal states with correct rewards, $T(s,a)$ the next state reached by taking action $a$ from state $s$, and $\mathcal{T}(s)$ the set of all possible trajectories from state $s$ to terminal states. Define two disjoint subsets of actions at state $s$, where $\mathcal{A}^+(s)=\{a \in \mathcal{A} | \exists \tau \in \mathcal{T}(T(s,a))\ {\rm such\ that}\ \tau\ {\rm ends\ in}\  \mathcal{C}\}$, and $\mathcal{A}^-(s)=\{a \in \mathcal{A} | \forall \tau \in \mathcal{T}(T(s,a)), \tau\ {\rm does\ not\ ends\ in}\ \mathcal{C}\}$. The intermediate rewards are given by a reasonable Process Reward Model (PRM) $r$. Terminal rewards $R$ indicate a correct solution ($R > 0$), and $0$ otherwise. A PRM is considered reasonable if the following conditions hold for all non-terminal states $s$: (i) $\forall a_{+}^{'} \in \mathcal{A}^+(s), \ \forall a_{-}^{'} \in \mathcal{A}^-(s)$: $r(s, a^-) < r(s, a^+)$, and (ii) $\forall a_{+} \in \mathcal{A}^+(s), \ \forall a_{-} \in \mathcal{A}^-(s), \ \forall \tau^- \in \mathcal{T}(T(s,a^-)), \ \forall \tau^+ \in \mathcal{T}(T(s,a^+))$: $\sum_{i \in \tau^-} r_i < \sum_{i \in \tau^+} r_i$. If these reasonable PRM conditions hold, then the greedy policy with respect to $Q^{\pi_u}$ defined by $\pi_{\rm greedy}(s) = \arg\max_a Q^{\pi_u}(s,a)$ always selects actions leading to a correct terminal state.
>
> This extension demonstrates that our novel theoretical framework and algorithm remain effective, as might be provided by process reward models.
>
> ### Why this matters: more efficient algorithms through structural insight
>
> By recognizing these inherent characteristics of LLM reasoning MDPs, we can develop more efficient algorithms. Traditional RL algorithms like PPO were designed for control tasks with fundamentally different structures summarized in Table 2 in our manuscript, where GRPO/DAPO can be seen as variants of PPO, which may
> - Carry unnecessary computational overhead (e.g., critic model)
> - Requires complex estimators for cyclic state spaces when simpler estimators may suffice for tree structures
> - Requires extensive tuning for stability for various implementation-level tricks, as shown in [2], where performance is highly sensitive to various implementation choices.
>
> Our approach exploits the natural structure of LLM reasoning tasks to achieve higher quality and diversity in an efficient manner. Our work opens the door to lightweight, theoretically-grounded implementations specifically designed for the expanding domain of LLM reasoning.
>
> In addition, our work also enriches classic RL by revealing unexpected findings: in traditional RL, uniform policies are often viewed as uninformative starting points rather than meaningful strategies. However, our theoretical analysis demonstrates that in deterministic tree-structured MDPs, the Q-function of the uniform random policy contains crucial information that can be directly exploited to find optimal solutions without complex generalized policy iteration methods.
>
> ### Additional empirical validation
>
> **Experiments with tool use.**
>
> To better address the reviewer's concern, **we conduct additional experiments to validate ROVER in tool-use scenarios based on open-sourced Search-R1 [1]'s setup**. Specifically, we fine-tune the Qwen2.5-3B-Base model on the combined training sets of Natural Questions (NQ) and HotpotQA. Given limited time and resources during the rebuttal phase, we evaluate ROVER trained with only 300 steps (lower than 500 steps of GRPO in the original Search-R1 paper [1]). However, ROVER outperforms the baseline by an obvious margin, which demonstrates its great potential in tool-use experiments.
>
> **(Performance comparison)** The performance across multiple benchmarks, spanning both General QA and Multi-hop QA, is presented in the table below, which **shows that ROVER not only works effectively with tools but actually outperforms the original GRPO in terms of average performance (even without extensive hyperparameter tuning of ROVER)**:
>
> | | NQ        | TriviaQA  | PopQA     | HotpotQA | 2wiki     | Musique   | Avg.      |
> | -- | -- | - | -| - | - | - | - |
> | Search-R1-base (GRPO) | 0.421     | 0.583     | 0.413     | 0.297    | 0.274     | 0.066     | 0.342     |
> | ROVER                 | **0.477** | **0.600** | **0.443** | 0.290    | **0.286** | **0.069** | **0.360** |

---

> > ### Author Response · Authors · 2025-11-21
> > **Response to Reviewer Qj2i (Part 3)**
> >
> > **(Diversity comparison)** To evaluate diversity, we generated 32 responses for each prompt from the Bamboogle benchmark (with the temperature set to 1.0). We then use the cosine distance between agent responses as the diversity metric. The results in the table below indicate that ROVER shows better diversity.
> >
> > |  Method| cosine distance (evaluated on Bamboogle) |
> > | :-- | :- |
> > | Search-R1-base (GRPO) | 0.042 |
> > | ROVER | **0.065**  |
> >
> > We argue that diversity is crucial for tool-use tasks, which also represents a potential future direction for our simple yet effective ROVER algorithm. An agent capable of generating diverse queries can retrieve more varied and comprehensive information from the corpus, thereby increasing its chances of discovering the key evidence required to correctly answer a prompt. Conversely, a model lacking diversity may fall into monotonous query patterns, leading to a failure loop where it repeatedly fetches the same unhelpful results from the search engine and is unable to solve the problem.
> >
> > ### Intermediate rewards and stochastic transition dynamics.
> >
> > Since widely-used RLVR benchmarks basically follow the deterministic dynamic transitions with a terminal reward that indicates the correctness, we modify the didactic MDP in Figure 5 in our paper to validate the practical effectiveness of ROVER for intermediate rewards and stochastic transitions.
> > - For scenarios considering intermediate rewards, we assign a reward of 1 to the intermediate states {$\{A,B,C,D,AC,BD,CA,DB\}$} that can lead to the correct terminal state. In this environment, ROVER still captures all optimal solutions with a 100\% success rate (detailed performance comparison is included in Appendix C.1).
> > - For stochastic transitions, we follow the widely-adopted sticky action concept [5] in classic RL literature to inject stochasticity in token-level transition dynamics. Specifically, we modify the environment dynamics to be stochastic by setting $P(s+a|s,a)=0.75, P(s+a'|s,a)=0.25$, where $a'$ denotes the action different from the chosen action $a$. This means that there is a 0.25 probability for the agent to travel a random path, thereby ensuring the dynamic stochastic. Our empirical results demonstrate that ROVER can effectively handle this kind of stochasticity. If the dynamic becomes deterministic at test time, ROVER successfully captures all modes with a 100\% success rate. If the dynamic remains stochastic at test time, ROVER achieves a 77\% success rate, which is still high considering the 25\% randomness in the environment. (detailed performance discussions are included in Appendix C.2).
> >
> > ### Conclusion
> >
> > Rather than imposing restrictions, our framework identifies and leverages the essential inherent structure of LLM reasoning tasks with verifiable rewards. By recognizing the fundamental differences between reasoning and classic RL domains, we enable more principled, lightweight implementations, enabling higher performance while maintaining diversity specifically tailored for LLM reasoning across a broad range of applications.
> >
> > > W2: While the paper claims robustness, the ablation shows performance is sensitive to the temperature parameter $\rho$. An extremely low $\rho$ causes collapse, while a high $\rho$ reduces to random sampling. The default $\rho=1$ works well here but might need tuning for other domains.
> >
> > Thanks for the insightful observation regarding the temperature parameter, which is the only parameter sensitive to performance. However, $\rho=1$ transfers well across all tasks and model sizes throughout our experiments, which is not sensitive to the specific content of reasoning tasks. Please note that the temperature parameter $\rho$ has a clear theoretical interpretation in our framework, which controls the trade-off between exploration and exploitation. And this is different from arbitrary hyperparameters in many RL algorithms that lack direct theoretical justification. Our approach demonstrates significantly greater robustness than alternative RL methods as analyzed in [2].
> >
> > While no method is completely parameter-free, our approach offers principled defaults with theoretical justification and demonstrates robust performance across a practical range of settings. Compared to alternative approaches requiring extensive hyperparameter tuning [2], our method represents a significant advance in robustness and practical applicability.

---

> > > ### Author Response · Authors · 2025-11-21
> > > **Response to Reviewer Qj2i (Part 4)**
> > >
> > > > Q1: Theorem 1 proves that acting greedily according to the random policy's Q-values is optimal. Why, then, does the practical ROVER algorithm use a softmax sampler instead of a purely greedy approach?
> > >
> > > In Section 3.1, we establish our random policy valuation framework, whose core components involve estimating the value of the random policy (policy evaluation), and choosing an action for execution based on the value function (action selection). As the reviewer pointed out, Theorem 1 demonstrates an interesting result: acting greedily according to the random policy's Q-values is indeed optimal. While sole greedy action selection based on random policy's Q-values guarantees solution quality, which is sufficient for traditional RL tasks (e.g., Atari games [5]), it sacrifices solution diversity, which is crucial for math reasoning, as there can be multiple viable response paths that yield correct solutions. According to our theoretical analysis, random policy's Q-values effectively characterize the probability of reaching successful outcomes following the selected action, where higher Q-values indicate action branches with denser successful pathways. Based on this insight, we propose a soft sampler for improving diversity while guaranteeing quality (analysis provided in Theorem 2, which characterizes the performance bound). Theorem 2 justifies the use of the softmax sampler to balance between exploiting these high-value pathways and maintaining necessary solution diversity.
> > >
> > > > Q2: How would this be helpful if extend to other more complicated domain than math?
> > >
> > > We have conducted additional experiments in LLM reasoning with tool use, following Search-R1's setup. Please see our detailed responses to W1.
> > >
> > > **Reference**
> > >
> > > [1] Jin, Bowen, et al. "Search-r1: Training llms to reason and leverage search engines with reinforcement learning." arXiv preprint arXiv:2503.09516 (2025).
> > >
> > > [2] Liu, Zihe, et al. "Part i: Tricks or traps? a deep dive into rl for llm reasoning." arXiv preprint arXiv:2508.08221 (2025).
> > >
> > > [3] Zhang, Kaiyan, et al. "A survey of reinforcement learning for large reasoning models." arXiv 2025.
> > >
> > > [4] Zeng, Weihao, et al. "Simplerl-zoo: Investigating and taming zero reinforcement learning for open base models in the wild." COLM 2025.
> > >
> > > [5] Machado, Marlos C., et al. "Revisiting the arcade learning environment: Evaluation protocols and open problems for general agents." Journal of Artificial Intelligence Research 61 (2018): 523-562.
> > >
> > > ---
> > >
> > > **We thank the reviewer for the time and effort in reviewing our work! We would very much appreciate it if the reviewer could check our responses and the updates in the paper and let us know whether they address your concerns. We are happy to provide further clarification if you have any additional concerns.**

---

> ### Author Response · Authors · 2025-11-26
> **Looking forward to your reply!**
>
> Dear Reviewer Qj2i,
>
> Considering there have been more than 5 days since we posted our response, we'd like to ask if we have addressed your concerns. We believe that we have clarified the concerns and questions raised.
>
> Specifically, we provided a new theorem and additional experiments to validate the generalizability of our method to new environments (`see Appendix A.3 and Appendix C`), new tool-use experiments to demonstrate the superior performance of our method on other domains (`see Appendix I`), and detailed clarifications on the temperature parameter $\rho$ and softmax operator.
>
> This will be a good improvement to our manuscript. We are happy to provide further clarification if you have any additional concerns. Thanks again for your feedback.
>
> Best,
>
> The authors

---

> > ### Comment · Reviewer_Qj2i · 2025-11-27
> >
> > Thank you, I will keep my positive rating.

---

> > > ### Author Response · Authors · 2025-11-28
> > >
> > > Thanks for the feedback and the positive assessment, and we greatly appreciate your time and effort for reviewing our work!

---

### Author Response · Authors · 2025-11-21
**General Response**

We thank all of the reviewers for their time and insightful comments. We are encouraged to find that reviewers generally recognize the novelty, theoretical grounding, and effectiveness of our work. Specifically, we highlight the following strengths acknowledged by the reviewers:

*   **Novel Methodological Perspective:** Reviewers commend ROVER for introducing a "fresh perspective" by framing inference-time optimization as random policy evaluation [Qj2i]. This is recognized as a principled and original approach that differs from standard RL paradigms [PNvP].
*   **Theoretical Depth:** The theoretical analysis is highlighted as a core contribution. Reviewers appreciate how our framework rigorously exploits the inherent properties (e.g., deterministic transitions, tree structures) of LLM reasoning to provide solid justification [Qj2i].
*   **Strong Empirical Performance:** Reviewers recognize the "notable gains" in output metrics compared to strong baselines like PPO and GRPO [Qj2i, 8GQr], confirming the method's effectiveness across diverse benchmarks.
*   **Diversity Preservation:** The ability of ROVER to maintain solution diversity while ensuring quality is acknowledged as a significant advantage [8GQr], addressing a critical challenge in current reasoning approaches.

We thank all the reviewers for their helpful and constructive feedback to improve the quality of our work. We have carefully updated our paper to incorporate the valuable suggestions. We summarize the revisions and added experiments as follows:

*   **[okcT] Comparisons with Stronger Baselines:** We conducted additional experiments comparing with **the suggested powerful implementation of DAPO** (under the setup of 20k maximum response length, significantly larger than 8k in our original manuscript) and evaluated on **Llama-3.1-8B**. ROVER maintains its performance advantage, confirming its effectiveness across different experimental configurations and model families. (`See Appendix H`)
*   **[Qj2i, 8GQr, PNvP] Experiments on Tool-Use:** We added extensive experiments following the **Search-R1** setup. Results demonstrate that ROVER consistently outperforms GRPO in tool-use scenarios (spanning General QA and Multi-hop QA) in terms of both average performance and response diversity. (`See Appendix I`)
*   **[Qj2i, 8GQr, PNvP] Theoretical Extensions & Robustness:** We extended our theoretical framework to accommodate **intermediate rewards** (Theorem 3 in Appendix A.3) and conducted ablation studies validating ROVER's robustness under **stochastic transitions** and **graph-structured MDPs**. (`See Appendix A.3 and Appendix C`)
*   **[PNvP] Comparison with Diversity-Seeking Methods:** We added a comparison with **GRPO with entropy regularization**. The results show that ROVER achieves a superior balance between performance improvement and diversity preservation. (`See Appendix G.3.3`)
*   **[okcT, PNvP] Clarifications:** We improved the clarity of the paper regarding the definition of "random policy valuation," the distinction from prior theoretical works, and implementation details (e.g., stop-gradient targets).

---

All changes in the new PDF are highlighted in **blue**.

We hope to have addressed all the raised concerns and would be happy to respond to further questions and suggestions.

---

### Author Response · Authors · 2025-11-30
**Letter to AC, SAC, and PC: Summarization of the discussion phase (Part 1)**

Dear AC, SAC, and PC,

We sincerely thank you for your time and effort in handling our paper! We would like to provide a summary of the key points from the reviews, our rebuttal, the feedback received, how we addressed the reviewers' concerns, and the consensus reached before the reset for your consideration.

**(Reviewers' recongnitions)** Reviewers acknowledge that our proposed ROVER algorithm provides a **novel** perspective by framing RL as random policy evaluation [Qj2i, PNvP], offering a **theoretically grounded** approach that leverages the inherent structure of LLM reasoning [Qj2i], where the experiments demonstrate that ROVER achieves **notable gains** in **both performance and diversity** compared to **strong baselines** like PPO/GRPO across multiple benchmarks and metrics [Qj2i, 8GQr, okcT, PNvP]. All the reviewers like the simplicity, computational efficiency, and reproducibility of our method, which is designed based on our surprising theoretical finding.

**(Summary of our rebuttal status)**
- Initial status: **The majority of reviewers were positive (initial rating: 6, 6, 6, 2).**
- Post-rebuttal: We have thoroughly addressed these concerns in our rebuttal and the revised manuscript (highlighted changes in blue). We believe our responses have effectively addressed reviewers' concerns, which results in a unanimous **6-6-6-6** consensus before the shuffle.
  - Notably, **reviewer okcT (initial rating: 2) explicitly acknowledged that the concerns were addressed by our new experiments and explanations for the theory, and raised the rating to 6 accordingly.**
  - Reviewer Qj2i maintained the positive rating.
  - Reviewer PNvP stated that our response "strengthened their view", and requested a minor follow-up clarification regarding configuration details. We provided the requested clarification, but the discussion period closed before the reviewer could offer a final confirmation.
  - Reviewer 8GQr did not provide further comments during the available discussion period.

**(Summary of reviewers' concerns and our rebuttal actions)** We summarize the main concerns raised by each reviewer and how our rebuttal addresses each of the concerns below (points confirmed by reviewers are marked with &#9745;):

(1) **Reviewer Qj2i** (rating: 6, confidence: 4): The reviewer acknowledged our additional experiments and theory, maintaining a positive rating.

&#9745; [W1, Q2] Generalizing to other MDP structures (stochastic transition dynamics, intermediate rewards, and tool-use):
- Theory: We **extended our theoretical framework to accommodate intermediate rewards** (Theorem 3 in Appendix A.3).
- Experiments: We **added additional experiments validating ROVER's robustness on other MDP structures**, including stochastic transition dynamics and intermediate rewards (Appendix C). We further added extensive experiments on **tool-use environment**, following Search-R1 [3] setup, where results demonstrate that ROVER consistently outperforms GRPO in tool-use scenarios (spanning General QA and Multi-hop QA) in **both average performance and response diversity** (Appendix I).

&#9745; [W2] Sensitivity to the temperature hyperparameter:
- We clarified the robustness of our method, highlighting that ROVER outperforms baselines across all experiments without parameter-tuning.

&#9745; [Q1] Softmax vs. greedy sampling:
- We provided additional justification of the theoretical support for the quality-diversity tradeoff with soft sampling.

(2) **Reviewer 8GQr** (rating: 6, confidence: 4): The reviewer raised 3 weaknesses (W1-3) and 4 questions (Q1-4), but did not respond during the available discussion period. We have fully addressed the concerns in the revision.

[W1, W2, Q2, Q3] Generalizing to other MDP structures (stochastic, intermediate rewards, graph-structured MDP, and more complex RLVR settings):
- Addressed via our new experiments with promising results on stochastic transitions, graph-structured MDPs, tool use (based on Search-R1 [3]), and theoretical extensions to accommodate intermediate rewards (Appendix A.3, C, I).

[W3] Results with 70B+ models.
- While 70B experiments were computationally infeasible, we have demonstrated ROVER's consistent robustness across different model families and scales (up to 8B), suggesting its generalizability.

[Q1, Q4] Computational/memory comparison, and failure/negative cases.
- We provided detailed comparisons on computational overhead and negative case analysis in our response.

---

> ### Author Response · Authors · 2025-12-03
> **Letter to AC, SAC, and PC: Summarization of the discussion phase (Part 2)**
>
> (3) **Reviewer okcT** (rating: 2 $\to$ 6, confidence: 3): The reviewer acknowledged our clarifications and new experiments, raising their score to 6.
>
> &#9745; [W1] Differences from a previous work [1]:
> - We clarified the detailed theoretical distinctions and the novelty of our paper compared to [1].
>
> &#9745; [W2] Empirical performance on settings of [2]:
> - We added additional experiments comparing with the suggested powerful implementation of DAPO [2] (under the setup of 20k maximum response length, significantly larger than 8k in our original manuscript), confirming our effectiveness across different experimental configurations.
>
> &#9745; [W3] Explain random policy evaluation and the performance on other models:
> - We added additional experiments on a totally new model, Llama-3.1-8B. ROVER maintains its performance advantage compared with baseliens, validating its effectiveness on different model families (`See Appendix H)`.
>
> (4) **Reviewer PNvP** (rating: 6, confidence: 4): The reviewer stated our response "strengthened their view", and asked a minor follow-up question just before the cutoff.
>
> &#9745; [W1, W3, Q2] Explanation of the details and novelty of practical implementation techniques:
> - We provided detailed clarifications regarding the design principle (Q function, loss function) and the novelty of our method.
>
> &#9745; [W2, Q3] Comparison with a diversity-aware baselines:
> - We added a comparison with a diversity-seeking baseline, GRPO with entropy regularization [4]. The results show that ROVER achieves a superior balance between performance improvement and diversity preservation. (`Appendix G.3.3`).
>
> [Q1] Generalizing to other MDP structures, including stochastic, intermediate-reward, or more complex tasks involving tool use.
> - Addressed via our new experiments with promising results on stochastic transitions, graph-structured MDPs, tool use (based on Search-R1 [3]), and theoretical extensions to accommodate intermediate rewards (`Appendix A.3, C, I`).
> - We provided further requested clarification on transition dynamics and the corresponding requirement of ROVER in Appendix I.
>
> **(Summary of the contributions)** To help AC better evaluate our paper, **we highlight our contributions as follows**:
> - **We prove a surprising result**. Theoretically, in the deterministic tree-structured MDPs with either binary terminal rewards or intermediate rewards that characterize LLM reasoning, the optimal policy can be derived directly from Q-values evaluated under a uniformly random policy. This is a finding that fundamentally simplifies RL for this domain.
> - Based on the theoretical foundation, we propose a practical and minimalist RL algorithm, ROVER, which is a simplified and robust framework compared to the current complicated RLVR methods, which maintains diversity during its reasoning process.
> - Despite ROVER’s radical simplification, extensive experiments
> across diverse tasks (including math reasoning and tool-use domain) and various models (including Qwen3-4B, Qwen3-8B, and Llama-3.1-8B) demonstrate that it consistently achieves superior performance in terms of both quality and diversity compared with previous SOTA methods (including standard and diversity-seeking baselines).
>
> We hope this summary helps clarify the status of the paper, and we also believe the unanimous consensus reached before the rollback reflects the solid quality of the work. We greatly appreciate your careful consideration of our responses and the overall review situation in your final evaluation.
>
> Best,
>
> The authors
>
> **Reference**
>
> [1] Laidlaw et al., Bridging Reinforcement Learning Theory and Practice with the Effective Horizon. NeurIPS 2023.
>
> [2] Wang et al., Beyond the 80/20 Rule: High-Entropy Minority Tokens Drive Effective Reinforcement Learning for LLM Reasoning. NeurIPS 2025.
>
> [3] Jin, Bowen, et al. "Search-r1: Training llms to reason and leverage search engines with reinforcement learning." COLM 2025.
>
> [4] Jiang, Yuxian, et al. "Rethinking entropy regularization in large reasoning models." arXiv preprint arXiv:2509.25133 (2025).

---

### Meta-Review · Area_Chair_DeTa · 2026-01-12

**Summary:**

This paper proposes ROVER, a minimalist reinforcement learning method for LLM reasoning that bypasses generalized policy iteration by showing that, under the deterministic tree-structured MDPs induced by autoregressive reasoning with verifiable rewards, the optimal policy can be recovered directly from the Q-function of a uniformly random policy. Across the initial reviews, three reviewers were clearly positive (scores of 6), while one reviewer was initially negative (2), citing concerns about theoretical scope, generalization, and empirical breadth. After the rebuttal and revision, the discussion converged to a unanimous positive consensus, with the initially negative reviewer explicitly raising their score to 6.

**Reviewer Concerns:**

1.	Narrow theoretical assumptions (deterministic, tree-structured MDPs; terminal binary rewards).
2.	Limited exploration of intermediate rewards, stochasticity, graph-structured MDPs, and tool-use scenarios.
3.	Sensitivity to the temperature hyperparameter and robustness.
4.	Scale of experiments and comparison to strong baselines.

**Reviewer Scores:**

Likely to be postive.

---

### Decision · Program_Chairs · 2026-01-26

Accept (Poster)